# Lumbar spine MRI annotation with intervertebral disc height and Pfirrmann grade predictions

**Friska Natalia[1], Sud Sudirman[2]\*, Daniel Ruslim[3], Ala Al-Kafri[4]**

**1** Information Systems Department, Universitas Multimedia Nusantara, Tangerang, Indonesia, **2** School of Computer Science and Mathematics, Liverpool John Moores University, Liverpool, United Kingdom, **3** Radiology Department, MRCCC Siloam Hospital, Jakarta, Indonesia, **4** School of Computing, Engineering and Digital Technologies, Teesside University, Middlesbrough, United Kingdom

\* s.sudirman@ljmu.ac.uk

**Data Availability Statement:** URL: https://data.mendeley.com/datasets/x6ggzp2ycn/1 DOI: 10.17632/x6ggzp2ycn.1.

**Funding:** Initials of the authors who received each award: FN Grant numbers awarded to FN: 004-RD-

## Abstract

Many lumbar spine diseases are caused by defects or degeneration of lumbar intervertebral discs (IVD) and are usually diagnosed through inspection of the patient's lumbar spine MRI. Efficient and accurate assessments of the lumbar spine are essential but a challenge due to the size of the clinical radiologist workforce not keeping pace with the demand for radiology services. In this paper, we present a methodology to automatically annotate lumbar spine IVDs with their height and degenerative state which is quantified using the Pfirrmann grading system. The method starts with semantic segmentation of a mid-sagittal MRI image into six distinct non-overlapping regions, including the IVD and vertebrae regions. Each IVD region is then located and assigned with its label. Using geometry, a line segment bisecting the IVD is determined and its Euclidean distance is used as the IVD height. We then extract an image feature, called self-similar color correlogram, from the nucleus of the IVD region as a representation of the region's spatial pixel intensity distribution. We then use the IVD height data and machine learning classification process to predict the Pfirrmann grade of the IVD. We considered five different deep learning networks and six different machine learning algorithms in our experiment and found the ResNet-50 model and Ensemble of Decision Trees classifier to be the combination that gives the best results. When tested using a dataset containing 515 MRI studies, we achieved a mean accuracy of 88.1%.

## 1. Introduction

Lumbar spine diseases are a major cause of disability and pain worldwide. Defects in intervertebral discs (IVDs), caused either by injuries or degenerative conditions, are responsible for a large proportion of spinal disorders [1]. One of the leading causes of lumbar spine diseases is lumbar spine stenosis, which is the narrowing of the lumbar spinal canal, where nerve roots exit the spine, due to defects in one of the IVDs. The narrowing of the lumbar spinal canal causes pressure on the nerve roots which then produces a wide range of symptoms associated with the disease [2]. Early diagnosis and treatment are essential for improving patient outcomes. However, its diagnosis can be challenging and time consuming, and requires trained

LPPM-UMN/ P-HD/VI/2022. The full name of funder: the Ministry of Education, Culture, Research, and Technology of the Republic of Indonesia URL of funder website: https://www.kemdikbud.go.id/ Did the sponsor or funder play any role in the study design, data collection and analysis, decision to publish, or preparation of the manuscript? No

**Competing interests:** The authors have declared that no competing interests exist.

radiologists because of a wide variation in imaging conditions that makes it difficult to detect important and relevant imaging features.

Unfortunately, there is a heavy demand for neuroradiologists and specialists in many countries around the world. The rapid increase in population size in many countries means that there is not a sufficient number of medical experts trained to provide a good level of service to the population. The National Health Service (NHS) in England reported a significant increase in the number of cases where the wait time for diagnostic radiology exceeds its maximum target of thirteen weeks [3]. The latest clinical radiology workforce census carried out in 2022 by the Royal College of Radiologists reported that the workforce is not keeping pace with demand for services (3% growth vs. 5% demand) and there is a 29% shortfall of clinical radiologists which is expected to worsen to 40% by 2027 [4]. This problem is expected to deteriorate further since the number of radiographical imaging including Magnetic Resonance Imaging (MRI) and Computed Tomography scans has always been increasing historically. This rationalizes the need for a new approach to increase the efficiency and effectiveness of the diagnostic radiology processes.

MRI is the standard modality for clinicians when inspecting a patient's lumbar spine to diagnose lumbar spine diseases. In addition to being radiation-free, the MRI imaging process allows for accurate interpretation of an IVD's condition through multi-view evaluation with good soft tissue contrast [5]. We previously put forward a strategy for assessing the extent of lumbar spinal stenosis in lumbar spine MRI by automatically calculating the anteroposterior diameter and foraminal widths of the last three lumbar spine IVDs [6]. The strategy starts with a semantic segmentation of T1- and T2-weighted composite axial MRI images using SegNet that partitions the image into six regions that include the three most important ones namely, the IVD, the Posterior Element, and the Thecal Sac. The accuracy of the segmentation results is improved by applying a novel contour evolution algorithm along the boundaries between the three important regions. Nine anatomical landmarks are then located on the image by delineating the region boundaries. The anteroposterior diameter and foraminal widths are determined with geometry using these nine landmark points. The algorithm's performance was evaluated on a dataset containing 515 MRI studies. The average error of the calculated right and left foraminal distances and the anteroposterior diameter relative to their expert-measured distances are 0.28 mm, 0.29 mm, and 0.90 mm, respectively.

The health of an IVD can be assessed by 1) analyzing its *nucleus pulposus* (or nucleus for short) compared with its *annulus fibrosus* (or annulus for short) on T2-weighted images, and 2) measuring its height as observed from the sagittal view. On MRI, the hyperintense signal of the nucleus on T2-weighted images has been shown to correlate directly with the health of the IVD and any reduction in intensity correlates with progressive degenerative changes [7]. Many clinicians use the Pfirrmann Grading system which is based on MRI signal intensity, disc structure, and distinctions among the nucleus, annulus, and disc height [8].

Deep learning is a very popular technique in medical imaging and has been proposed for the detection, classification, and grading of lumbar spine IVDs. This is because the technique allows learning directly using raw pixel data without manual feature engineering or explicit translation of expert knowledge into the algorithm. The technique has been shown to generalize well to new and unseen data provided sufficient images are used for the model development. An example of such an approach was proposed in [9]. The method uses a deep convolutional neural network to predict the Pfirrmann grade of 1000 T2-weighted sagittal images of 515 patients. The method uses a Yolov5 model [10] to detect the IVD regions in the image which will then assign one of the five Pfirrmann grades to each of the detected IVD. The accuracy of the classification is reported to be 95%.

The problems with similar techniques in the literature are threefold. The first is there is a lack of explicit localization of each IVD type (e.g., L5/S1, L4/L5, etc.). This information is important when informing clinicians on where the problem occurs. The second is there is no segmentation of the IVD regions thus making shape analysis of the IVD and the measurement of the IVD height impossible. Several approaches attempt to address each of these problems individually, including two by Alomari et al. for IVD localization [11] and herniation diagnosis [12]. The closest approach in the literature that attempts to address all the problems together is by Zheng et al. [13] who propose a deep learning-based technique, called the BianqueNet, to segment and locate each IVD, measure the IVD height, and quantify the degeneration factor by assigning one of eight modified Pfirrmann grades [14] using pixel intensity histogram and geometric features.

In this paper, we propose a novel method to annotate each individual lumbar spine IVD in a mid-sagittal image with its predicted height and Pfirrmann grade using a combination of deep learning-based image segmentation, geometry of the segmented IVD region, and machine learning classification using spatial image features derived from the pixels in the IVD region. We focused on the last three lumbar spine IVDs (L3/L4, L4/L5, and L5/S1) because they are more prone to injury than others. These IVDs bear the most weight and pressure from the upper body as well as higher twisting pressure due to the wider range of motion they need to support making it more susceptible to twisting and bending forces that can damage the discs [15, 16].

## 2. Materials and methods

Our proposed method for lumbar spine annotation involves IVD height measurement and Pfirrmann grade prediction. The method uses a deep learning technique to segment different regions of a mid-sagittal image, locate and identify each IVD, analyze the segmented IVD region geometrically to locate certain points in the region and calculate their distance as the IVD height, and extract image features from the region and use machine learning to predict the Pfirrmann grade. Fig 1 shows an overview of the proposed methodology. The manual IVD height measurement and degeneration assessment are carried out by an expert radiologist with over ten years of experience. This stage of the methodology aims to produce ground truth IVD height and Pfirrmann grade data to be used to evaluate the performance of the methodology later.

Each stage of the methodology will be described in this section, but first, we will start by describing the dataset used.

### 2.1 The dataset

The material used in this study was retrieved on 13 February 2023 from an archived sample of DICOM lumbar spine MRI images available publicly in [17]. This dataset contains 515 anonymized clinical MRI studies of patients with symptomatic back pains. We can confirm that the data does not contain any information that could be used to identify individual participants. The dataset consists of 48,345 T1-weighted and T2-weighted MRI images of each patient's lumbar spine taken using a 1.5-Tesla Siemens Magnetom Essenza MRI scanner and saved in the Digital Imaging and Communications in Medicine (DICOM) format. The following information is obtained by extracting the information stored in the metadata of the DICOM files. The scanning sequence used in all scans is Spin Echo (SE), which is produced by pairs of radio-frequency pulses, with segmented k-space (SK), spoiled (SP), and oversampling phase (OSP) sequence variants. Fat-Sat pulses were applied just before the start of each imaging sequence to

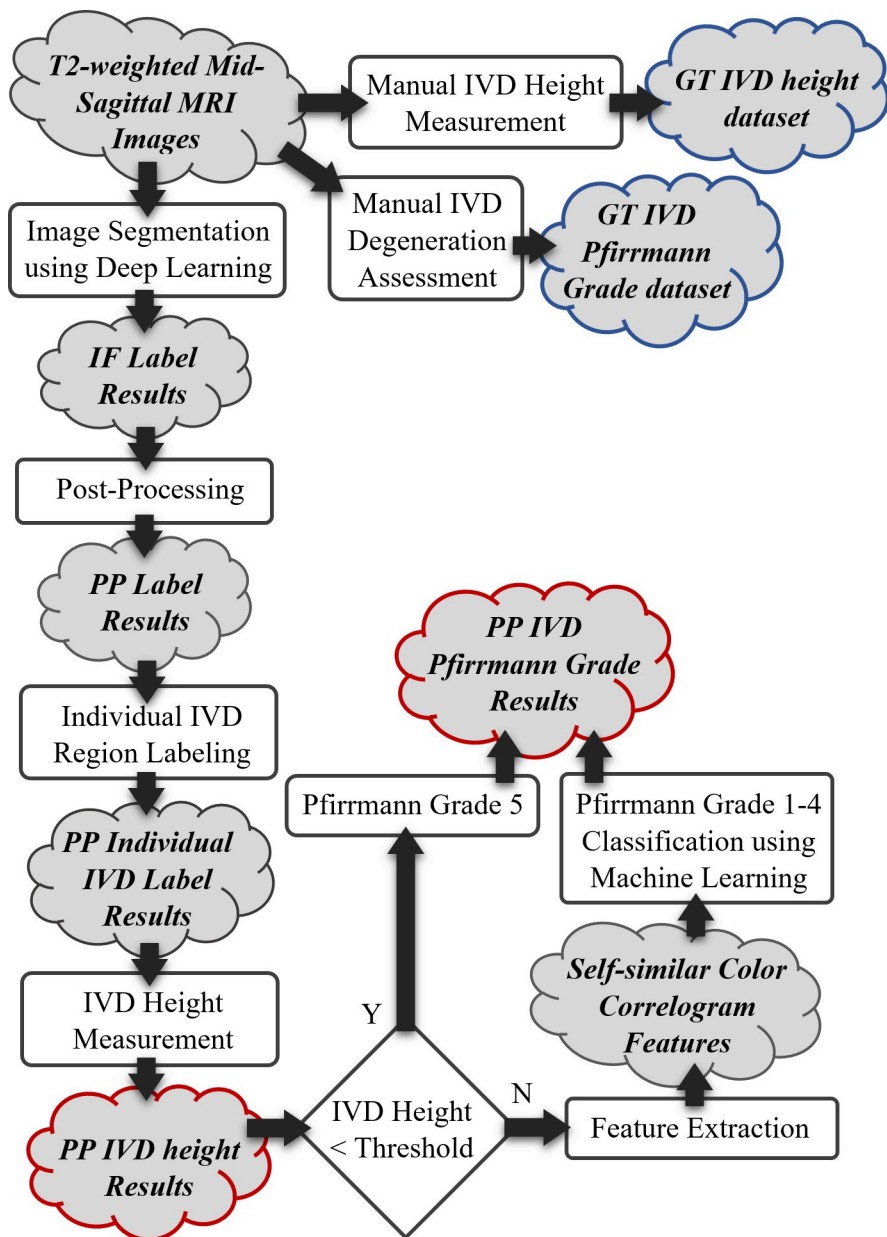

**Fig 1. A flowchart describing an overview of the methodology.** The red highlight marks the automated process output whereas the blue highlight marks the manual process output.

saturate the signal from fat matters to make it appear distinct from water. The range of acquisition parameter values used during sagittal MRI scans is provided in Table 1.

Each MRI study includes at least the last three lumbar vertebrae (L3, L4, and L5) and their adjacent posterior elements, the topmost sacral bones (S1), and the last three IVDs (L5/S1, L4/L5, and L3/L4). Each study contains both sagittal and traverse view scans and their corresponding cross-view information. This allows us to see the direction and position of the image plane of a traverse view slice on the sagittal view, and vice versa in a DICOM viewer application. Each study has at least four sequences, or sets of images, that correspond to a

**Table 1. MRI scanning parameters for T2-weighted sagittal scans.**

| Scanning Parameters | Value or Range of Values |
|---|---|
| Number of Echoes (ETL) | 1 |
| Repetition Time (milliseconds) | 3190 to 3660 |
| Echo Time (milliseconds) | 90.0 to 96.0 |
| Slice Thickness (mm) | 4 |
| Spacing Between Slices (mm) | 4.8 to 5.4 |
| Imaging Frequency (MHz) | 63.7 |
| Scanning Sequence | SE |
| Sequence Variant | SK\SP\OSP |
| Scan Options | SAT1 |
| Pixel Spacing X and Y (mm/pixel) | 0.6771 to 0.875 |
| Image Dimension X and Y (pixels) | 320 or 384 |
| Echo Train Length | 9, 13, 15, or 16 |
| Percent Sampling | 70 to 78 |
| Pixel Bandwidth | 150 to 195 |
| Flip Angle | 150 |

combination of T1-weighted or T2-weighted and sagittal or traverse scans. Each sequence is stored as a set of.ima files in a folder with the following folder naming format <T1/T2>_TSE_<SAG/TRA>_PatientID_SequenceID. For example, a folder named T2_TSE_-SAG_0001_0003 contains the images in the third sequence of the T2-weighted sagittal view study of the first patient.

## 2.2. Data selection and pre-processing

In each MRI study, a mid-sagittal image is automatically selected from the T2-weighted sagittal sequence using the image classification technique we described in [18] resulting in 515 T2-weighted mid-sagittal slices. Mid-sagittal images are defined as the image that is located closest to the median plane bisecting the body vertically through the midline roughly equally from the left and right side. These 515 images will be the input to the methodology we are describing in this paper. An expert radiologist examined the last three lumbar IVDs in each image to measure their heights and determine their condition by assigning a Pfirrmann grade to each, resulting in 1545 Pfirrmann grades. The distribution of the grades is 60, 1002, 167, 295, and 21 for Pfirrmann grades 1, 2, 3, 4, and 5, respectively. This information is stored as the ground truth data and to be used later to measure the accuracy of the proposed methodology.

All 515 mid-sagittal images are in Siemens DICOM image format. The Pixel Spacing (*ps*) attribute and the Image Dimension (*dm*) attribute of each DICOM image are extracted and stored as metadata. Although not all images have the same image dimension or pixel spacing, our observation of the data found that all images have an identical number of pixels and pixel spacing in both x and y-axis directions, therefore a single variable can be used to represent each attribute of both axes. Most of the images, 508 out of 515, have a pixel spacing of 0.7292, two images have a pixel spacing of 0.6771, one image has a pixel spacing of 0.7344, and four images have a pixel spacing of 0.8750. All pixel spacing has a unit of millimeters per pixel. The majority of the images, 511 of 515, have an image dimension of 384 x 384, and four images have a dimension of 320 x 320. We make the image dimension uniform across the dataset by scaling those having 320 x 320 dimensions to 384 x 384 using the cubic interpolation

technique. For each of these images, a new pixel spacing ($ps'$) is calculated by scaling the original pixel spacing with the dimension ratio, i.e.,

$$ps'_i = ps_i \times \frac{dm_i}{384} \tag{1}$$

Where $ps_i$, $ps'_i$, and $dm_i$ are the original pixel spacing, new pixel spacing, and original dimension of the $i^{th}$ image in the dataset. The number 384 comes from the new image dimension of the scaled image. After this, all images are then converted to greyscale and saved as PNG files.

## 2.3. Image segmentation using deep learning

The sequence of processes in this stage of the methodology is illustrated in Fig 2. This stage starts with manual segmentation of the MRI images to produce the Ground Truth (GT) label image dataset, followed by splitting the MRI and label image dataset into Training and Test datasets, with the former being used to train a deep learning segmentation model. The following is the description of the manual segmentation step.

With the help of the radiologist and three experienced image labelers, each MRI image is segmented into six distinct non-overlapping regions, namely Anterior, PosteriorA, PosteriorB, Vertebrae, IVD, and Sacrum. An example of an MRI image superimposed with color-coded labels of the six regions is shown in Fig 3.

A set of conditions is imposed when labeling each image to minimize the occurrence of labeling errors that could be introduced during the process. Each label image must satisfy the following six requirements:

1. There should be only one large Anterior region. The region must not have any holes.

2. There should be only one large PosteriorA region. The region must not have any holes.

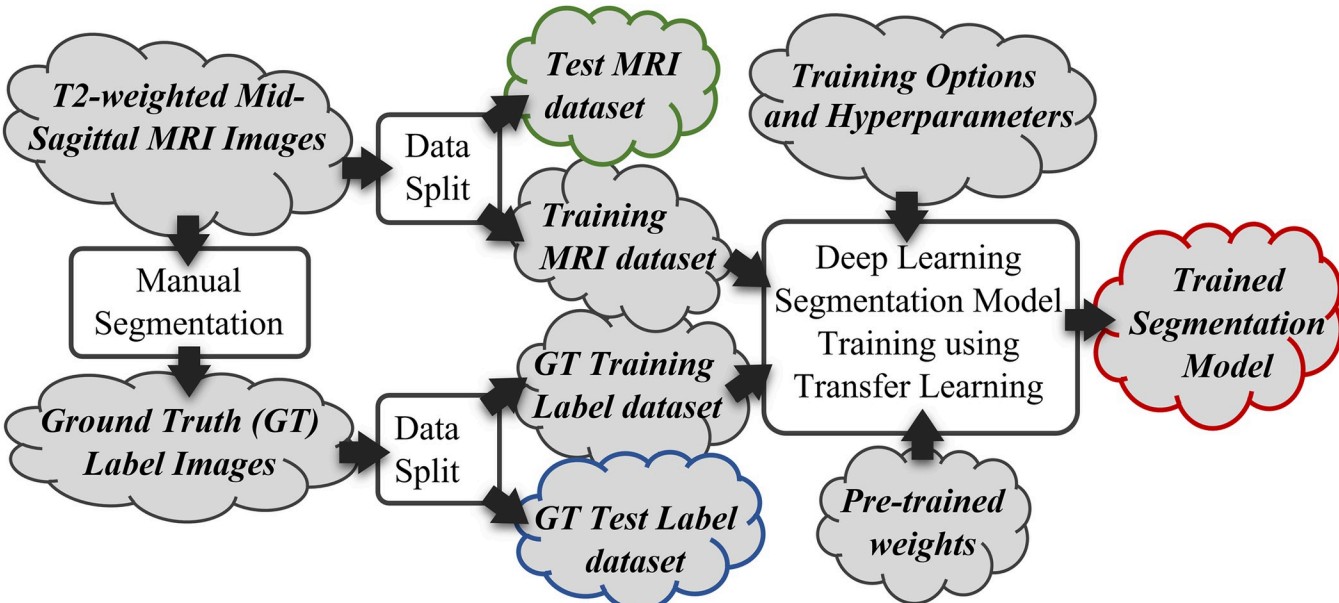

**Fig 2. The process to train the deep learning segmentation model.** The red, green, and blue-highlighted components are used subsequently in the performance analysis step, illustrated in Fig 4.

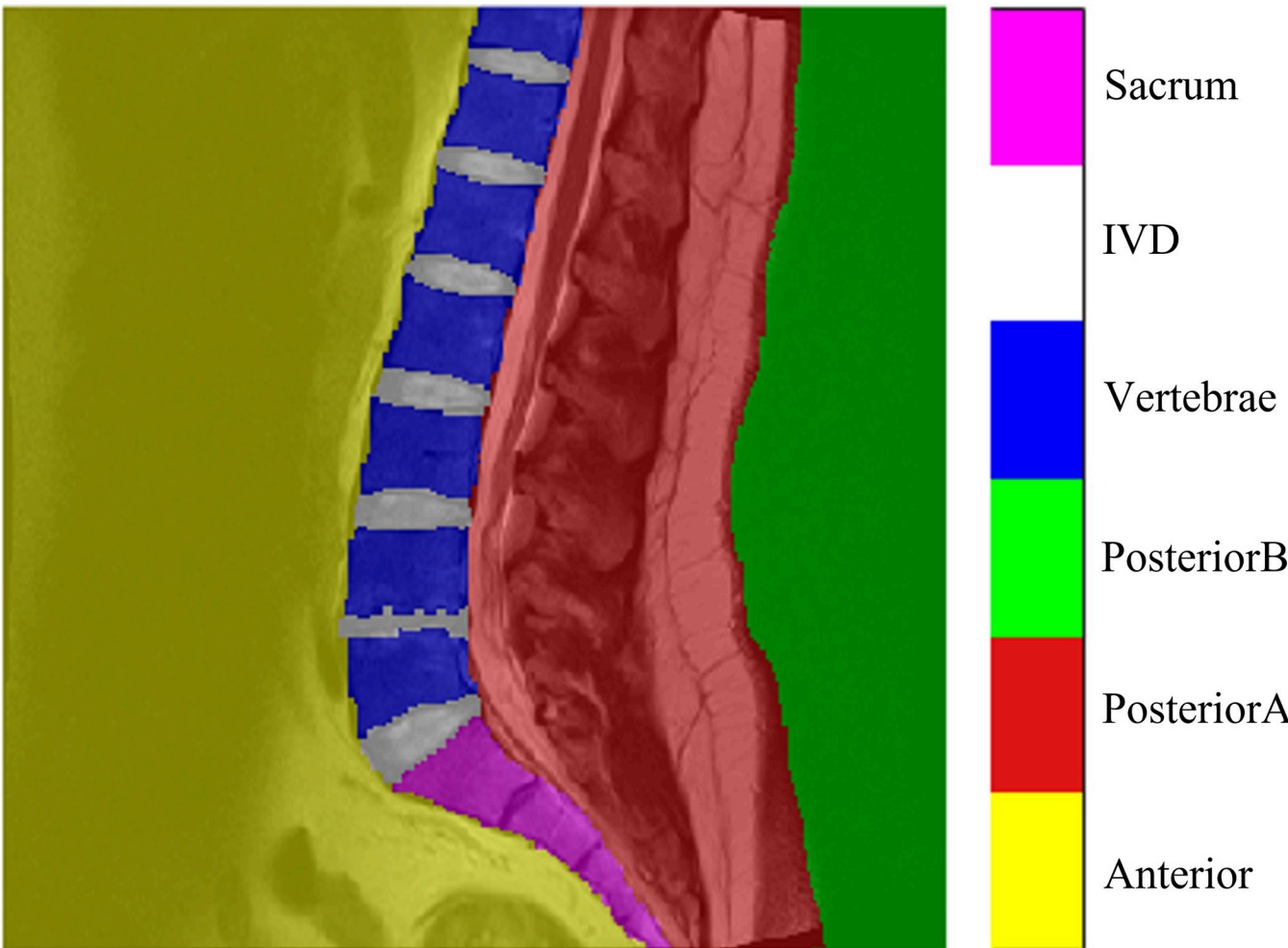

**Fig 3. An MRI image superimposed with color-coded labels of the six regions illustrating the location of each region in the image.**

3. There should be only one large Sacrum region. The region must not have any holes.

4. There is no requirement to have only one PosteriorB region but every PosteriorB region must not have any holes.

5. There should only be one large Vert-IVD region. The Vert-IVD region is defined as the union of all IVD and Vertebrae regions. The Vert-IVD region must not have any holes.

6. The relative placement of IVD and Vertebrae regions inside the Vert-IVD region is alternating, starting with an IVD region at the bottom (just above the Sacrum region), followed by a Vertebrae region, then another IVD region, and so on.

The data splitting step divides the MRI and label dataset into training and testing sets with an 80:20 ratio. The training sets are further divided by a 60:20 ratio to allow the smaller portion of the dataset to be used to validate the segmentation model during the training process. Our methodology uses, as the architecture for the model, the encoder-decoder structure with an atrous separable convolution known as DeepLabv3 architecture [19]. Five DeepLabv3 deep learning networks are considered, they are ResNet-18, ResNet-50 [20], MobileNetv2 [21],

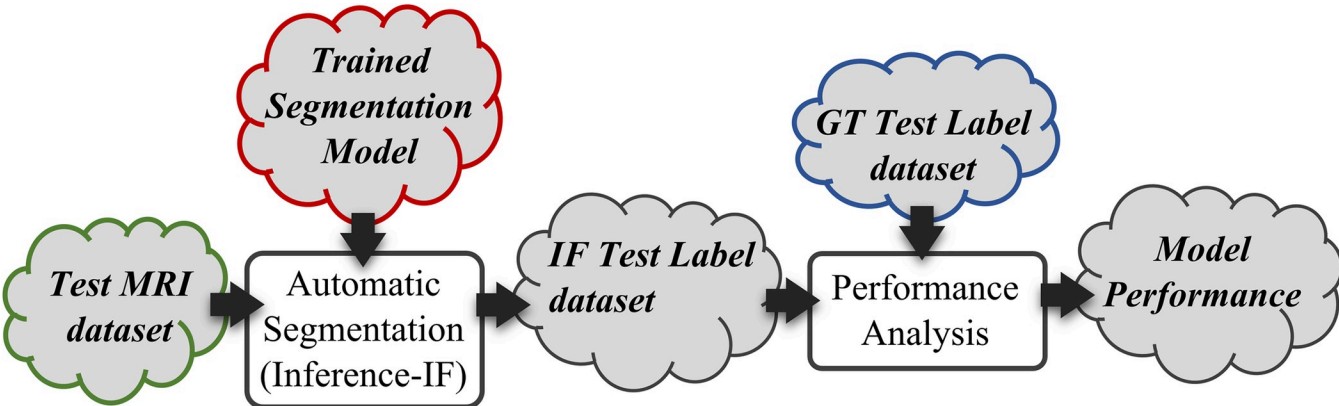

**Fig 4. The process to analyze the performance of the trained model using the test dataset.** The red, green, and blue-highlighted components are the same highlighted output components in the previous model training step, illustrated in Fig 2.

Xception [22], and InceptionResNetv2 [23]. We adopted the Transfer Learning approach when training the model by using pre-trained network weights as the model's starting weight values and using a small learning rate to tune the model to our training data. The performance of the model is then analyzed by using the model to perform inference on all images in the Test MRI dataset to create the Inferred Test label images and compare them with the Ground Truth Test label images. This step is illustrated in Fig 4.

The training process also considers several combinations of training options and hyper-parameter values to find the best-performing model for each network. The best overall model is chosen from all the trained models based on the average of the mean accuracy, intersection-over-union, and BFScore of the Vertebrae and IVD regions only. This model is then used to segment all the images in the MRI dataset to create the IF label dataset, to be used as the input to the next, Post-Processing, stage.

## 2.4. Post-processing

This stage is performed to ensure that the automatically segmented label images in the IF label dataset meet the same requirements as the manually created label images. This stage uses binary image processing and morphological techniques [24] to find Connected Component Regions (CCR) of the different region types in a label image and close all the holes in the CCRs. The steps in this stage are described below:

1. Find all CCRs of the detected Anterior region. If there is more than one CCR, keep the largest CCR and merge the smaller CCRs with the surrounding region. We merge smaller CCRs with the surrounding region because there can only be one CCR of the Anterior region in the entire image. We then proceed to find and close all holes in the largest CCR.

2. Repeat Step 1 for the PosteriorA and Sacrum regions individually.

3. Find all CCRs of the PosteriorB region and close any holes in them. We do not merge smaller CCRs with surrounding regions because our observation shows that there can be more than one PosteriorB CCR in a label image.

4. Merge the Vertebrae and IVD regions to make a combined region called the Vert-IVD region. Repeat Step 1 for the Vert-IVD region. The rationale is, that the Vert-IVD region should make up the main contiguous spine region, with Sacrum as a separate region at the

bottom. We ensure that any CCR marked as Vertebrae or IVD that are detached from this main Vert-IVD region are removed and the main Vert-IVD region is free of holes.

5. We then check the CCR of each Vertebrae and IVD region to make sure that they are not too small, which most likely occurs due to imperfection in the semantic segmentation, by counting the number of pixels in each CCR and merging them with the surrounding region if the number of pixels is smaller than 20.

We call the set containing the output images of this stage as PP label dataset which is to be used as the input to the next, Individual IVD Region Labeling, stage.

## 2.5. Individual IVD region labeling

Although by this stage we already have the information of all detected IVD regions, we have yet to determine the order or the sequence of the IVD CCR along the spine in order to assign a specific label to each IVD CCR. To achieve this, first, we need to find a sequence of points that thread along the middle of the spine connecting the bottom part to the top part of the spine. The idea is, by traversing along the line from the bottom up, we can ascertain the order of the IVD and Vertebrae CCRs as they are encountered and hence be able to label them in sequence, e.g., L5, L4, L3, L2, and L1 for the Vertebrae and L5/S1, L4/L5, L3/L4, L2/L3, and L1/L2 for the IVD. We always traverse from the bottom since it is always guaranteed to start from the Sacrum region. To find this sequence of points first we merge all IVD, Vertebrae, and Sacrum CCRs into one contiguous region. We then apply the morphological thinning technique [8] to find the thinnest middle line of this region. For brevity, we refer to this line as the spine line. The spine line will have a width of one pixel and run along the length of the region connecting the Sacrum CCR at the bottom to the topmost IVD or Vertebrae CCR.

The result of the morphological thinning process is a binary image with white foreground pixels marking the locations of the spline line. The process does not necessarily produce a single line with only two endpoints, because we could also end up with a line that has branches with false endpoints along the main line. These branches can occur at any point along the line, as illustrated in Fig 5.

The next step is to determine the sequence of coordinates of the points, denoted as $p_i \in \Re^2$ where $i$ denotes the order of point $p$ in the sequence, connecting the Start and Finish end points. We set the end point with the highest y-coordinate value (i.e., the lowest point) as the Start end point and set the end point with the lowest y-coordinate value (i.e., the topmost point) as the Finish end point. We then apply the A-Star pathfinding technique [9] on the binary image to get the sequence. The A-Star pathfinding technique is a popular technique used in the game development community to find the shortest path between two points given a grid containing passable and obstruction cells. In our case, the two points are the Start and Finish end points, the grid is the binary image, and the passable and obstruction cells are the white and black pixels, respectively. Any points that are not found to belong to $p_i$ are then removed. We would like to note that given there is only one unique path connecting the Start and Finish end points, using any other pathfinding algorithms is an equally valid alternative and should produce the same sequence of points. Once the sequence of coordinates has been established, we traverse the line from start to finish and count the number of times an IVD region is encountered. The first IVD region is assigned with the L5/S1 label, the second with the L4/L5 label, and so on. We do the same with the Vertebrae region. The first Vertebrae region is assigned with the L5 label, the second with L4, and so on.

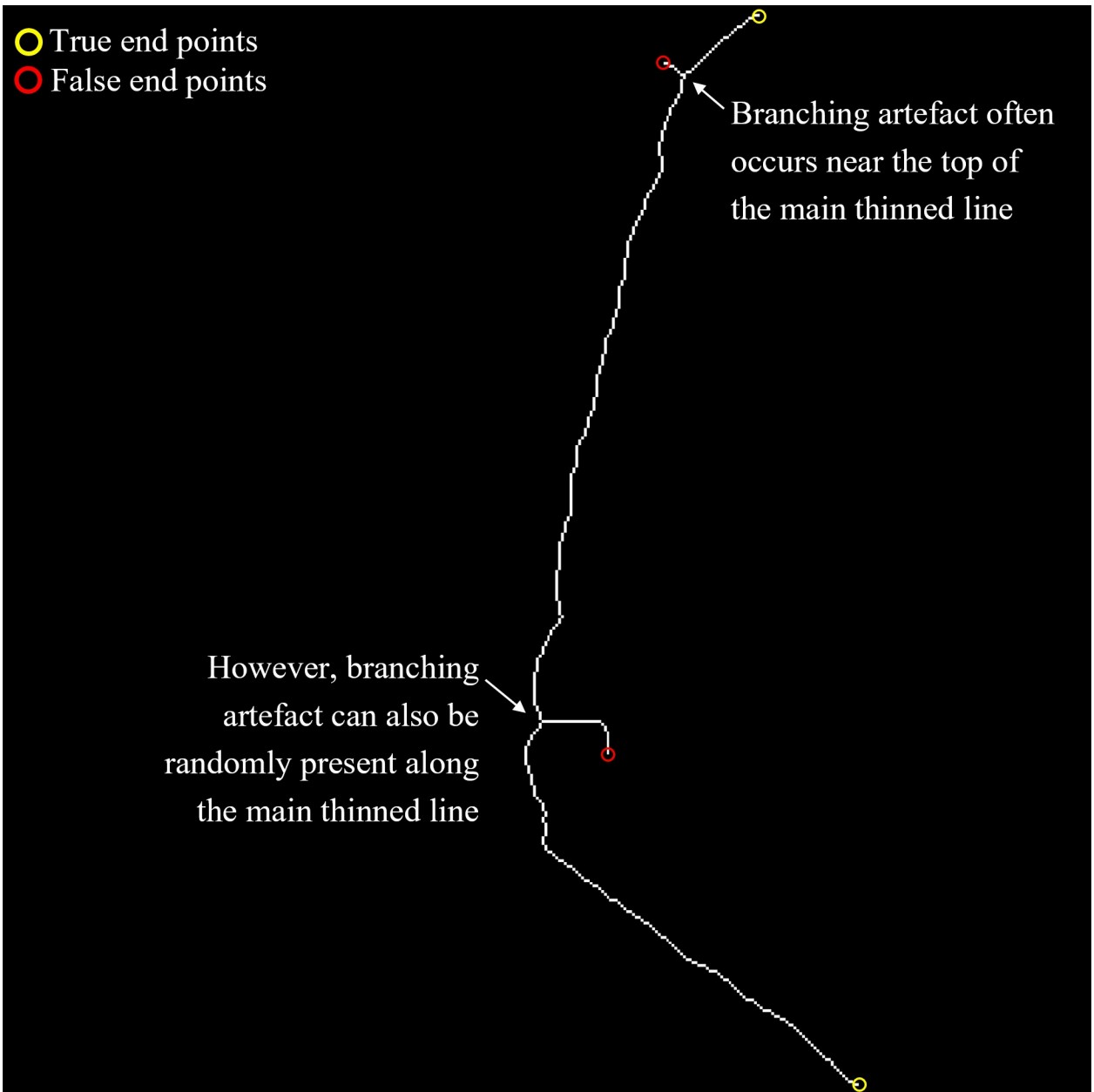

**Fig 5. An example result of a morphological thinning process, creating a spine line connecting the Sacrum part of the spine to the top part.** Two branches are apparent in this result showing two false end points in addition to the two true end points.

## 2.6. IVD height measurement

Finding the height of an IVD is essentially a process of finding a line segment that best bisects the IVD approximately parallel to the spine line. If the nucleus of the IVD is apparent, the line should connect the top and bottom edges of the nucleus, otherwise, its two end points should be about one millimeter above and below the bottom and top edges of the IVD, respectively. This process is a two-phase process that is performed on each detected IVD region from the previous stage. The first phase is the determination of $M_T$ and $M_B$ points, the two end points of

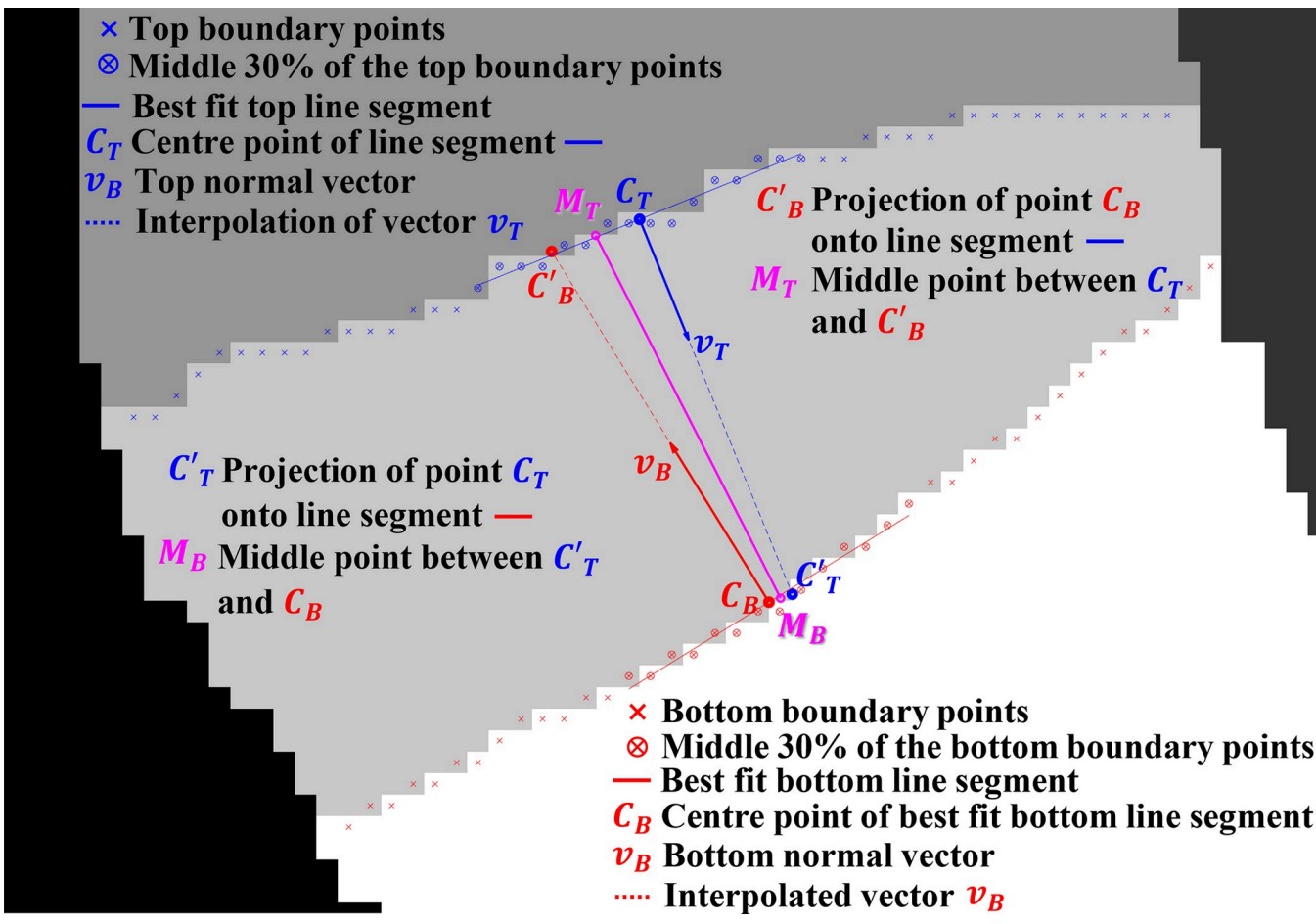

**Fig 6. Illustration of the different components of geometry used to determine the first estimate (magenta line) of the bisecting line segment of an IVD.**

the first estimate of the bisecting line segment. The steps involve geometry and are described using Fig 6 to illustrate the different components of the analysis, as follows:

1. Apply a low-pass Gaussian filter noise to reduce the amount of noise in the MRI image.

2. Upscale the smoothed MRI image by a factor of four using cubic interpolation. Also, upscale its corresponding label image by a factor of four using nearest interpolation.

3. Locate the top and bottom boundary points of the IVD. For example, for the L5/S1 IVD, the bottom boundary points mark the edges between the IVD and Sacrum regions and the top boundary points mark the edges between the IVD and L5 vertebrae.

4. Find the coordinates of the middle 30% portion of each boundary point and calculate the average coordinate of each middle portion to get the coordinates of $C_T$ and $C_B$, the top and bottom center points, respectively.

5. Apply linear regression to each middle portion to get the top and bottom best-fit lines.

6. Calculate vector $v_T$, which is perpendicular to the top best-fit line, and another vector $v_B$, which is perpendicular to the bottom best-fit line.

7. Extend vector $v_T$ from point $C_T$ to intersect with the bottom best-fit line to get point $C'_T$. Likewise, extend vector $v_B$ from point $C_B$ to intersect with the bottom best-fit line to get point $C'_B$

8. Calculate the coordinate of point $M_T$ as the halfway point between $C_T$ and $C'_B$. Similarly, calculate the coordinate of point $M_B$ as the halfway point between $C_B$ and $C'_T$.

The second phase refines the location of $M_T$ and $M_B$ points to get $M'_T$ and $M'_B$ points to make them closer to the edge of the IVD nucleus. We do so by searching the pixels in the vicinity of the initial points and in the direction of $v_T$ and $v_B$, respectively which has the highest pixel gradient. The search range is controlled by variable $k$, the higher the value of $k$ the longer the search range will be. The steps start by calculating the gradient magnitude and direction of each pixel in the MRI image that belongs to the IVD region using the Prewitt gradient operator. We denote the magnitude and direction of the gradient at pixel $p$ as $m_p$ and $\beta_p$, respectively. We also ensure that the range of $\beta_p$ is $-180° < \beta_p \leq 180°$. The steps to find the refined bottom point $M'_B$ are as follows:

1. Calculate vector $u_B$ which is the unit vector of $v_B$.

2. Calculate the vector's angle $\alpha$ as $\alpha = \tan^{-1}(v_{By}/v_{Bx})$ and $180 < \alpha \leq 180$, where $v_{Bx}$ and $v_{By}$ are the $x$ and $y$ components of vector $v_B$, respectively.

3. Calculate the coordinate of two points $A$ and $B$, where $A = M_B - u_B$ and $B = M_B + k \times u_B$, where $k$ is the search range variable.

4. For every pixel $p$ that lies on line segment $AB$, calculate the angle difference, $\delta_p$, at point $p$ as $\delta_p = \beta_p - \alpha$.

5. Multiply the magnitude of the pixel gradient with the cosine of the angle difference to get $b_p = \cos(\delta_p) \times m_p$.

6. The refined location of the bottom point $M'_B$ is determined as the point $p$ that has the highest value of $b_p$.

A similar step is used to find $M'_T$, but with using vector $v_T$ instead of $v_B$. The height of the IVD (in mm) is then calculated as the product of the Euclidian distance between $M'_T$ and $M'_B$ points and $ps'$, the current image's pixel spacing, calculated using Eq 1 and divided by four, the scaling factor.

## 2.7. Pfirrmann grade prediction

The process of assessing the degeneration of an IVD is akin to assessing the brightness and homogeneity of the pixels in the nucleus of the IVD and taking into account the height of the IVD. The brighter the pixels and the more homogenous the nucleus, the better the condition of the IVD. Our prediction algorithm of Pfirrmann grade is preceded by checking if the IVD height as calculated in the previous stage is shorter than a threshold. If it is, then the Pfirrmann grade is automatically set to 5. The height threshold is set to 3.0 mm. This value is based on the assessment of the radiologist on the data that we possess while making little assumption on what is considered as healthy or normal IVD height. Our review of the literature [25–27] in this regard yields quite wildly varying statistics on what is considered normal IVD heights, which could perhaps be attributed to the difference in race, genetics, and overall health of the population where the studies were conducted.

The Pfirrmann grade classification for IVD whose height is longer than the threshold takes into account the brightness and homogeneity of the pixels in the nucleus of the IVD and is

performed using Machine Learning, which requires extraction of image features from corresponding IVD region of the MRI image. The feature engineering component of this stage is described next.

**2.7.1. Feature engineering.** Before we extract any features from the MRI image, we first adjust the overall brightness of the MRI image according to the mean pixel intensity of the vertebrae regions. The rationale for this is that, when radiologists assess an IVD they do not use absolute brightness of the nucleus but rather its brightness relative to the surrounding vertebrae. Due to variations during the MRI scanning phase, the overall brightness of the MRI images can vary which reflects on the brightness of the vertebrae and IVD regions. An overexposed image will have a higher average pixel intensity than a normal one, which itself has a higher average pixel intensity than an underexposed one. To provide proof of this, we show the probability distribution of vertebrae pixel intensities over the entire dataset in Fig 7. In the figure, it can be seen that the pixel intensities have roughly a normal probability distribution with a mean value grey level value, $gl_{vert\_mean}$, of 69. In the absence of any widespread major defects in all of the vertebrae regions, this wide variation in the vertebrae pixel intensities can be attributed to the variation during the scanning stage.

When adjusting the brightness of an MRI image, we first calculate the mean intensity of all vertebrae pixels in the current image, denoted as $gl_{vert}$. We then subtract the difference between $gl_{vert}$ and $gl_{vert\_mean}$ and subtract the result from the original pixel intensities, $gl_{orig}$ to obtain the adjusted pixel intensity value, $gl$ as expressed in Eq 2.

$$gl = gl_{orig} - (gl_{vert} - gl_{vert\_mean}) \tag{2}$$

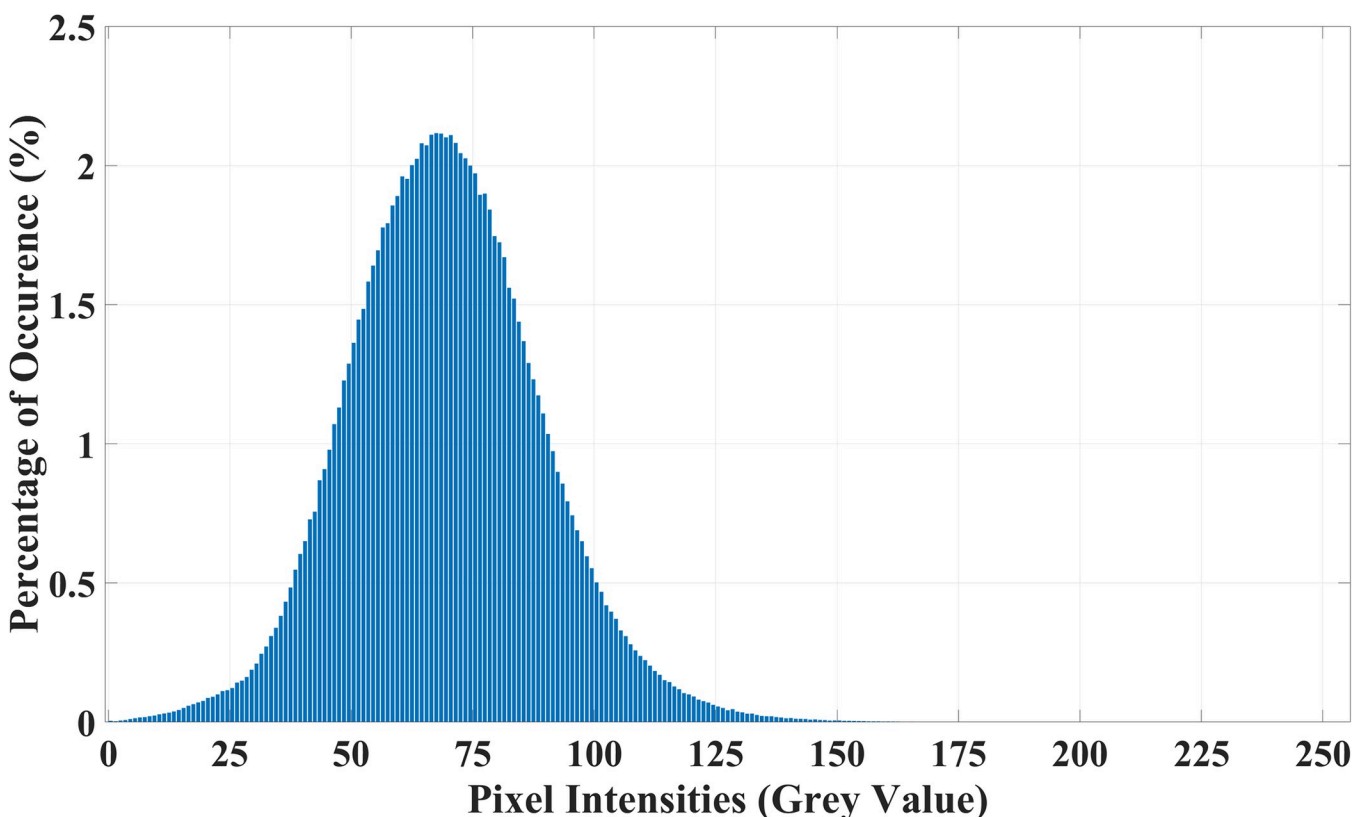

**Fig 7. The probability distribution of the pixel intensities belonging to all vertebrae regions in the entire dataset.**

We cap the adjusted pixel intensity values to between 0 and 255. It is worth emphasizing that the value of $gl_{vert}$ is calculated as the average pixel values in the Vertebrae regions in the current image whereas the value of $gl_{vert\_mean}$ is calculated as the mean of all $gl_{vert}$ values across the entire dataset. Therefore, by subtracting the bracket term from the original pixel value, we are offsetting the pixel value with the difference. If the current image has $gl_{vert}$ higher than $gl_{vert\_mean}$ that suggests the image is overexposed hence the pixel values will be reduced, and vice versa.

To capture the homogeneity of a region, we decided to use the color correlogram [28]. The color correlogram is a three-dimensional table indexed by color and distance between pixels. It expresses how the spatial correlation of color changes with distance in a stored image. It is a feature used in image analysis for indexing and comparing images. It captures the spatial correlation of colors and is effective for content-based image retrieval and pattern recognition, and has been found to be superior to other non-spatial features such as color histograms [29]. This is because the latter only describes the global color distribution in an image whereas the color correlogram includes spatial correlation information. It is robust to changes in appearance and shape caused by variations in viewing positions, camera zooms, and other imaging variables. Although color correlogram is a three-dimensional table when multiple distances are considered, in our case, we only use one distance hence our color correlogram is only a two-dimensional table. Also in our case, the color is an indexed greyscale value, $gl_{ind}$, produced by capping and quantizing the greyscale pixel intensities, as opposed to the literal meaning of the word color in the image processing context. The greyscale quantization and greyscale cap values are set to 30 and 150, respectively. The formula to derive the indexed greyscale value, $gl_{ind}$, from the adjusted pixel intensity, $gl$, given the greyscale quantization, $n_{gl}$, and greyscale cap $gl_{max}$, values are:

$$gl_{ind} = \left[\frac{\min(gl, gl_{max})}{\Delta_{gl}}\right] \tag{3}$$

$$\Delta_{gl} = \frac{gl_{max}}{n_{gl}} \tag{4}$$

Where $n_{gl}$ and $gl_{max}$ in our case were set to 30 and 150, respectively. The square bracket operator in Eq 3 denotes the floor() operator.

It is important to note that we do not calculate the color correlogram from the entire detected IVD region, but instead from its inner nucleus region. The step to detect the nucleus of an IVD starts by applying morphological erosion to the IVD region to shrink the size by 20%. This is to remove any non-IVD pixels belonging to neighboring regions that are incorrectly assigned as that IVD region. Then we identify candidate pixels by adaptive binary thresholding to the shrunk region. We start with a high value of the threshold, if the percentage of the number of pixels above the threshold is lower than 30% of the shrunk IVD region, we reduce the threshold and repeat the process until we do. The result at this point is our candidate nucleus region which may be fragmented. Then we locate the largest CCR of the candidate nucleus region and apply morphological closing to it. If the percentage of the number of pixels of the CCR is less than 20% of the shrunk IVD region, we repeat the process by reducing the threshold value. We will eventually get either a large contiguous CCR that is brighter than the rest of the IVD or the initial eroded IVD. The latter will occur if the IVD region has very low pixel intensities and the nucleus is not distinctly distinguishable from the annulus. If the nucleus CCR is detected, we then apply a further morphological erosion to it to shrink the size

by 10%, to get the inner nucleus region. It is from this region that we calculate the color correlogram. The image scaling factor, which is four, is used as the distance.

The feature that we extracted is a subset of the entire color correlogram. A color correlogram of $N$ number of colors has $N^2$ number of cells. The majority of information in a color correlogram is concentrated along the main diagonal cells of the matrix hence several studies have suggested not to use the entire matrix but only these cells [30, 31]. They are called auto color correlograms. We modified this approach by also using cells in the region around the main diagonal cells as our image feature. We call this feature, the *self-similar color correlogram* feature. The width of the region, denoted as $w$, determines how large the region is. Fig 8 illustrates a color correlogram with the number of colors $N = 10$. The yellow-shaded cells are the main diagonal cells and the blue-shaded cells are cells within the region width $w = 2$. The length of the feature, denoted as $len$, can be expressed as a function of $N$ and $w$:

$$len = N(2w + 1) - w(w + 1) \tag{5}$$

The values of $N$ and $w$ should always be positive integer numbers.

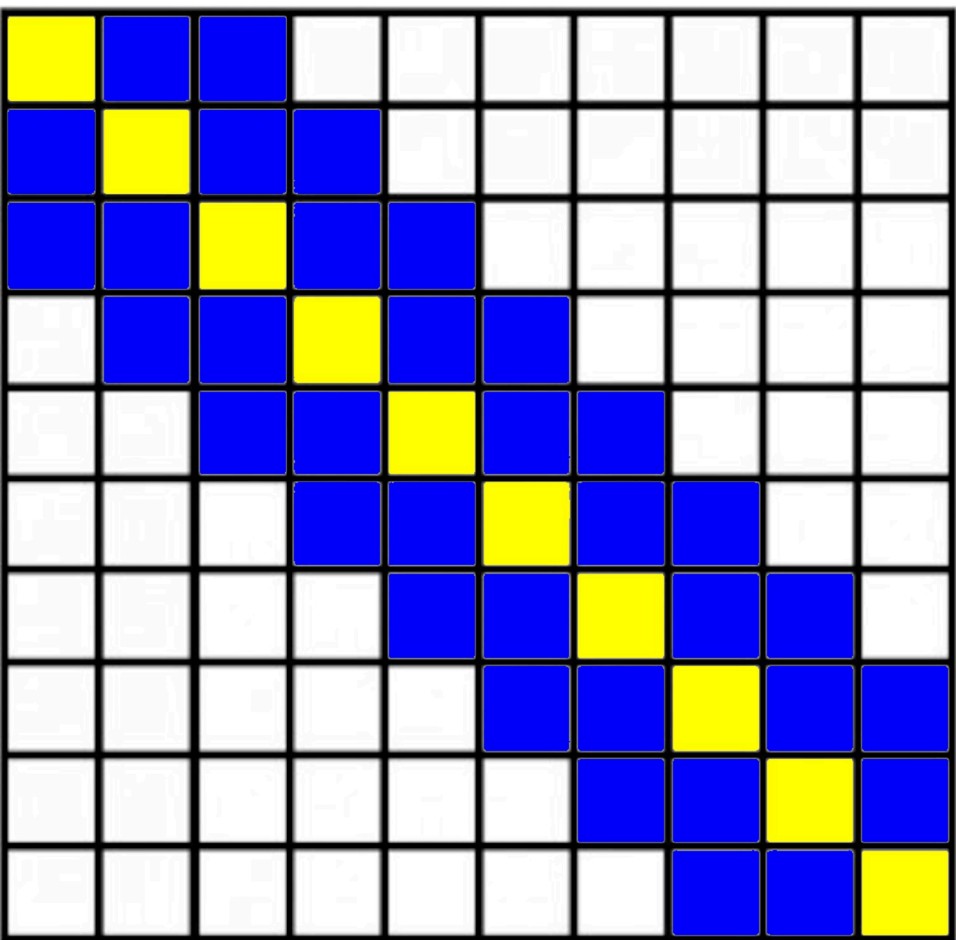

**Fig 8. An illustration of the color correlogram of ten unique colors.** Each cell in a matrix, Cij, contains the probability of occurrence of color i and j between two pixels, d distance apart. The yellow-shaded cells are the auto color correlogram, where i is identical to j, whereas the blue-shaded cells represent additional information included in our self-similar color correlogram feature where |i-j|<w and w = 2.

**2.7.2. Machine learning classification.** The classification of grades 1, 2, 3, and 4 of the Pfirrmann grade is performed by machine learning using the self-similar color correlogram feature. The training of the machine learning classifier uses ground truth data consisting of the actual Pfirrmann grade annotated by the radiologist (GT IVD Pfirrmann Grade dataset), the actual IVD height measured by the radiologist (GT IVD Height dataset), and the color correlogram feature calculated using the manually segmented label image (GT Label Image dataset). Six machine learning classifiers are considered, they are K-Nearest Neighbors, Classification Tree, AdaBoostM2 Classification Tree, Discriminant Analysis, Feedforward Fully Connected Neural Network [32], and Error-Correcting Output Codes [33]. The best model of each classifier type is obtained by incorporating 10-fold cross-validation during training. This means the models are trained 10 times using ten unique non-repeating training sets consisting of 90% of the data and tested with the remaining 10%. The final trained models are then used to predict the Pfirrmann grade of the same set of MRI images using IVD heights and color correlogram features calculated using the automatically segmented label image (PP Label Image dataset).

## 3. Results and discussion

In this section, we will present the implementation of the methodology followed by the experimental results and their analysis. The code was implemented using MATLAB on a PC running Microsoft Windows 11 operating system with an i9-13900K Intel CPU, two NVidia GeForce RTX 4090 GPUs, and 128 GB RAM.

### 3.1. Image segmentation and labeling results

The implementation of the image segmentation stage using the five DeepLabv3 deep learning networks, namely ResNet-18, ResNet-50, MobileNetv2, Xception, and InceptionResNetv2, is presented here. Each network was used to produce multiple models trained using a range of combinations of training options and hyperparameter values. The parameters are optimization algorithm, initial learning rate, learning rate drop factor, learning rate drop period, L2-regularization factor, gradient decay factor, and number of epochs. The performance of the models is measured using three metrics namely accuracy, intersection over union, and BFScore. Since we consider the vertebrae and IVD regions as the most important regions, we therefore only consider the segmentation performance of those regions. We use the mean score, calculated as the average of the six performance metrics, as an indication of which model to use for the next stage.

The performance of the best model for each network type over the search range of hyperparameter values and training options is provided in Table 2. As shown in the table, ResNet-50

**Table 2. The performance of the best model for each network type.**

|  | ResNet-18 | ResNet-50 | Mobile NetV2 | Xception | Inception ResNetV2 |
|---|---|---|---|---|---|
| **Vertebrae** |  |  |  |  |  |
| Accuracy | 97.0 | 97.2 | 96.8 | 97.0 | 97.1 |
| IoU | 92.0 | 92.4 | 91.5 | 91.5 | 92.3 |
| BFScore | 99.3 | 99.5 | 99.2 | 99.0 | 99.5 |
| **IVD** |  |  |  |  |  |
| Accuracy | 97.7 | 97.7 | 97.5 | 97.6 | 97.7 |
| IoU | 87.6 | 88.2 | 87.5 | 87.4 | 88.0 |
| BFScore | 98.9 | 99.1 | 99.1 | 98.8 | 99.3 |
| **Mean Score** | 95.4 | **95.7** | 95.3 | 95.2 | **95.7** |

**Table 3. The values of hyperparameters and training options that were used to train the best model.**

| Hyperparameters/training options | Value |
| --- | --- |
| Optimization Algorithm | ADAM |
| Initial Learning Rate | $10^{-3}$ |
| Learning Rate Drop Factor | 0.3 |
| Learning Rate Drop Period | 10 |
| L2 Regularization Factor | 0.005 |
| Gradient Decay Factor | 0.9 |
| Number of Epochs | 120 |
| Mini Batch Size | 16 |

and InceptionResNetV2 models have an identical mean score of 95.7. We opted to use the ResNet-50 model since it is faster to train, 47 minutes compared to the 88 minutes required for InceptionResNetV2.

The values of hyperparameters and training options that were used to train the best ResNet-50 model are provided in Table 3.

The trained ResNet-50 model was then used to segment the entire MRI image dataset to produce the IF label image dataset. Each image in this dataset was then post-processed to produce the PP label image dataset. This dataset is then used to detect and label each individual IVD and vertebrae. The results have 100% accuracy in terms of correct label names assigned to each IVD and vertebrae in all cases notwithstanding the performance of the segmentation stage before that. An example of the vertebrae and IVD labeling results are shown in Figs 9 and 10, respectively.

### 3.2. IVD height measurement results

Fig 11 shows a visualization of the IVD height measurement stage. The red semi-transparent region highlights the L5/S1 IVD and the small green semi-transparent region highlights part of the L4/L5 IVD above it. The two magenta points mark the initial top and bottom, $M_T$ and $M_B$, endpoints. The yellow points mark the refined top and bottom end points, $M'_T$ and $M'_B$, of the bisecting line segment obtained after shifting the original points in the directions of the blue and red arrows, respectively. The yellow dotted line is the final bisecting line segment from which the IVD height is calculated.

As for the overall performance, the predicted IVD heights were compared with the actual IVD height for the entire image in the dataset. We set the relative difference between the predicted and actual IVD height, defined as the former minus the latter, as the accuracy measure of the IVD height measurement stage. Their statistics are shown as box charts in Fig 12. The figure shows that the first, second, and third quartiles of the data are well within one millimeter. There are, however, quite a few outliers in the results and some are very severe, including one case where the predicted height is 12.8 mm longer than the actual height.

We noted five outlier cases and analyzed each case of the outliers to ascertain the causes and found that the majority of them are associated with trauma affecting the top cartilaginous end plate of the vertebrae below the disc as exemplified in Fig 13. During the manual IVD height measurement, the radiologist took the vertebrae defect into account and estimated the line where the missing cartilaginous end plate of the vertebrae would have been. Since they look similar to IVD and the two regions are often connected, the deep learning segmentation model always considers them as one IVD region. This error is then propagated to the IVD height measurement stage causing the algorithm to produce the unusually high value of output.

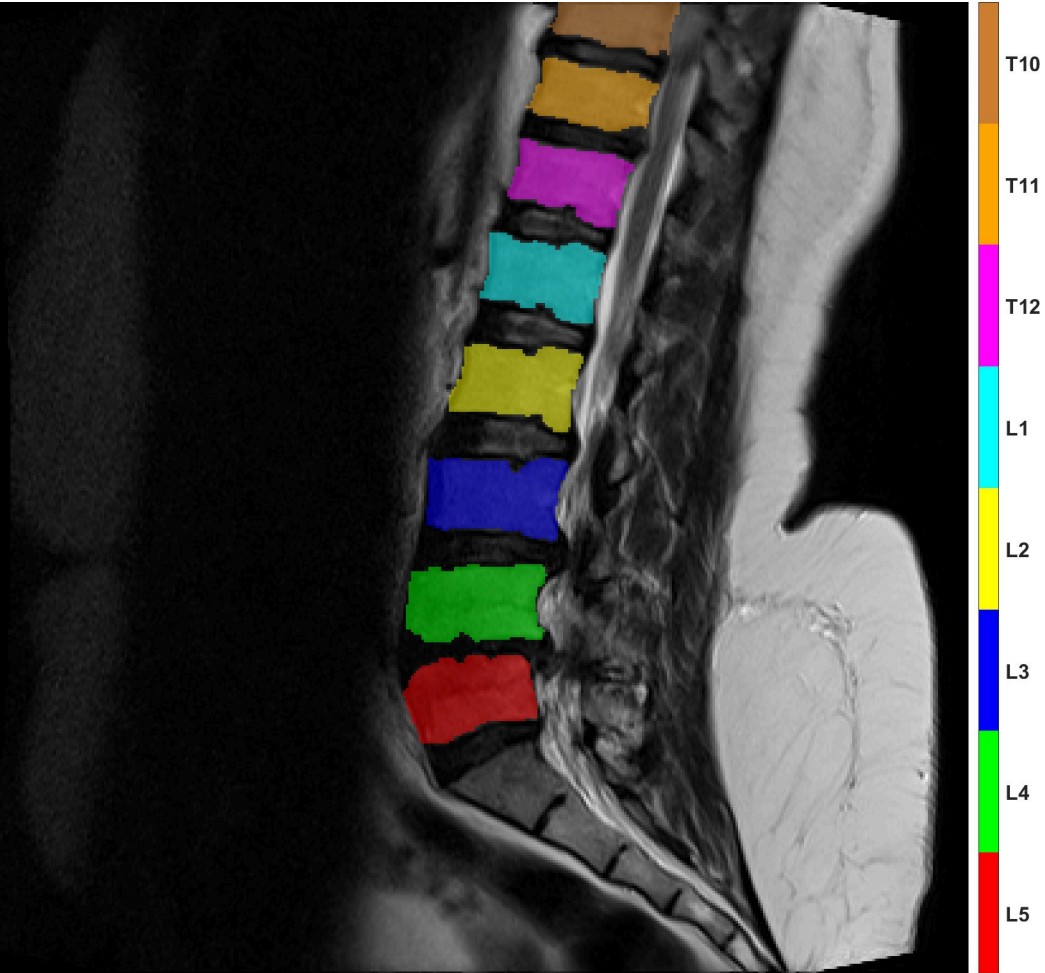

**Fig 9. An example of the vertebrae regions labeling result.**

### 3.3. Pfirrmann grade prediction results

Following the IVD height measurement stage, the Pfirrmann grade prediction stage is implemented as a combination of the height function and machine learning classification using the self-similar color correlogram features. The implementation considered six machine learning classifiers, which are K-Nearest Neighbors (KNN) with Error-Correcting Output Codes (ECOC) [33], Support Vector Machine (SVM) with ECOC, Discriminant Analysis (DA) with ECOC, Feedforward Fully Connected Neural Network, Decision Tree, and Ensemble of Decision Trees. Before the training, the best training options were selected using Hyperparameter optimization. The training also uses custom weight values for each data based on its class frequency. The class training weight $tw_c$ of class $c$ is calculated as:

$$tw_c = \frac{\text{median}(f)}{f_c} \tag{6}$$

Where $f$ is a set containing the frequency of occurrence of each class, and $f_c \in f$, is the frequency of occurrence of class $c$.

The best training options were then used to train the best model for each classifier type. The performance of the models was then evaluated using a 10-fold cross-validation technique.

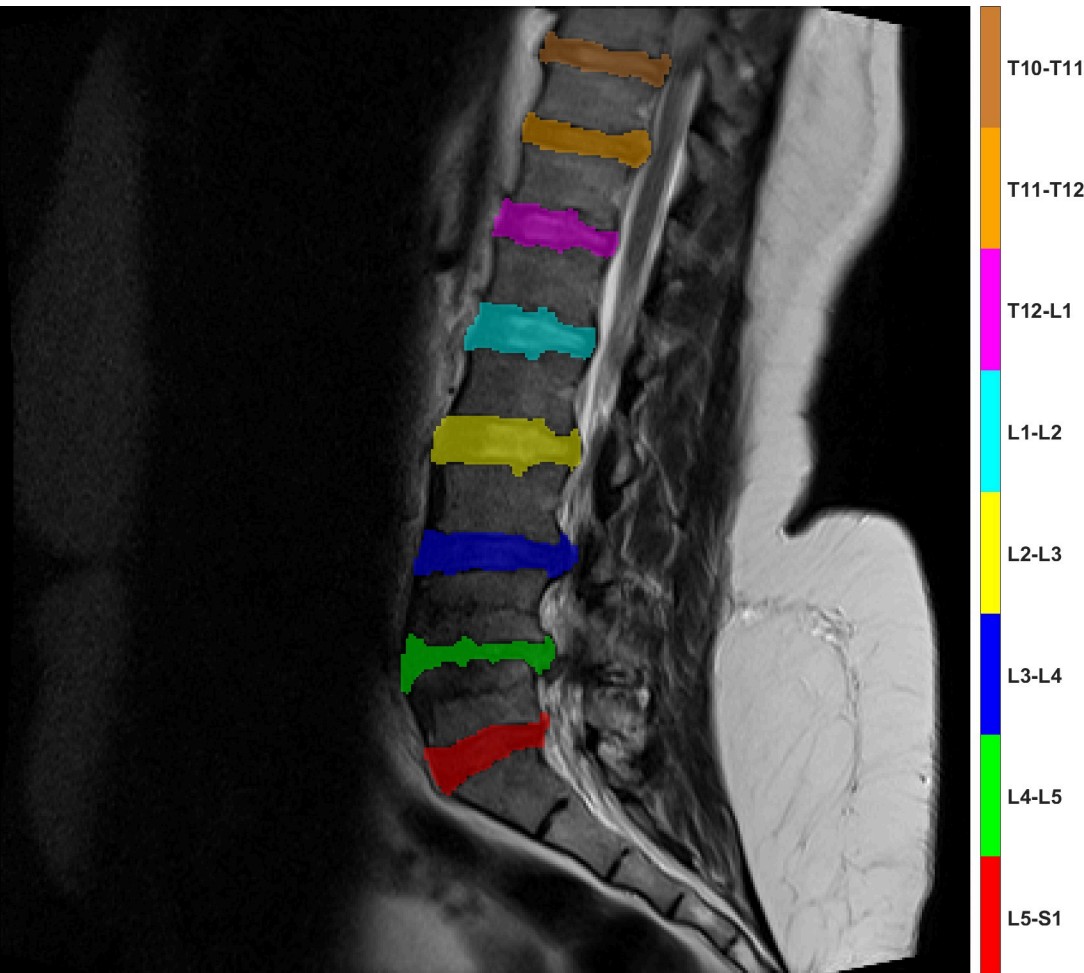

**Fig 10. An example of the IVD regions labeling result.**

This means ten versions of the same model of each classifier type were developed, each trained using 90% of the data and evaluated using the remaining 10%. The metric used to assess the model performance is the mean accuracy, which calculates the percentage of correctly predicted Pfirrmann grades in the entire dataset.

The performance of the models is given in Table 4. The table indicates that all classifiers produce relatively comparable results ranging from 78.2% using the DA with ECOC classifier to 88.1% using the Ensemble of Decision Trees classifier. The resulting Ensemble of Decision Trees classifier model is then used as the chosen model in the subsequent analysis. The search range and the best value of the hyperparameters are given in Table 5.

### 3.4. Analysis, discussion, and future work

The result of the combined Pfirrmann grade prediction approach using the best Ensemble of Decision Trees classifier and the predicted IVD height is shown as a confusion matrix in Fig 14. Note that the overall or mean accuracy of this classifier is 88.1% as mentioned in Table 4. That figure is the average of the class accuracies weighted by the percentage population of each class. If we dissect further the Pfirrmann grade classification results in Fig 14, we will find that the class accuracies for Pfirrmann Grade 1 to 5 are 68.3%, 92.6%, 79.0%, 82.7%, and 76.2%,

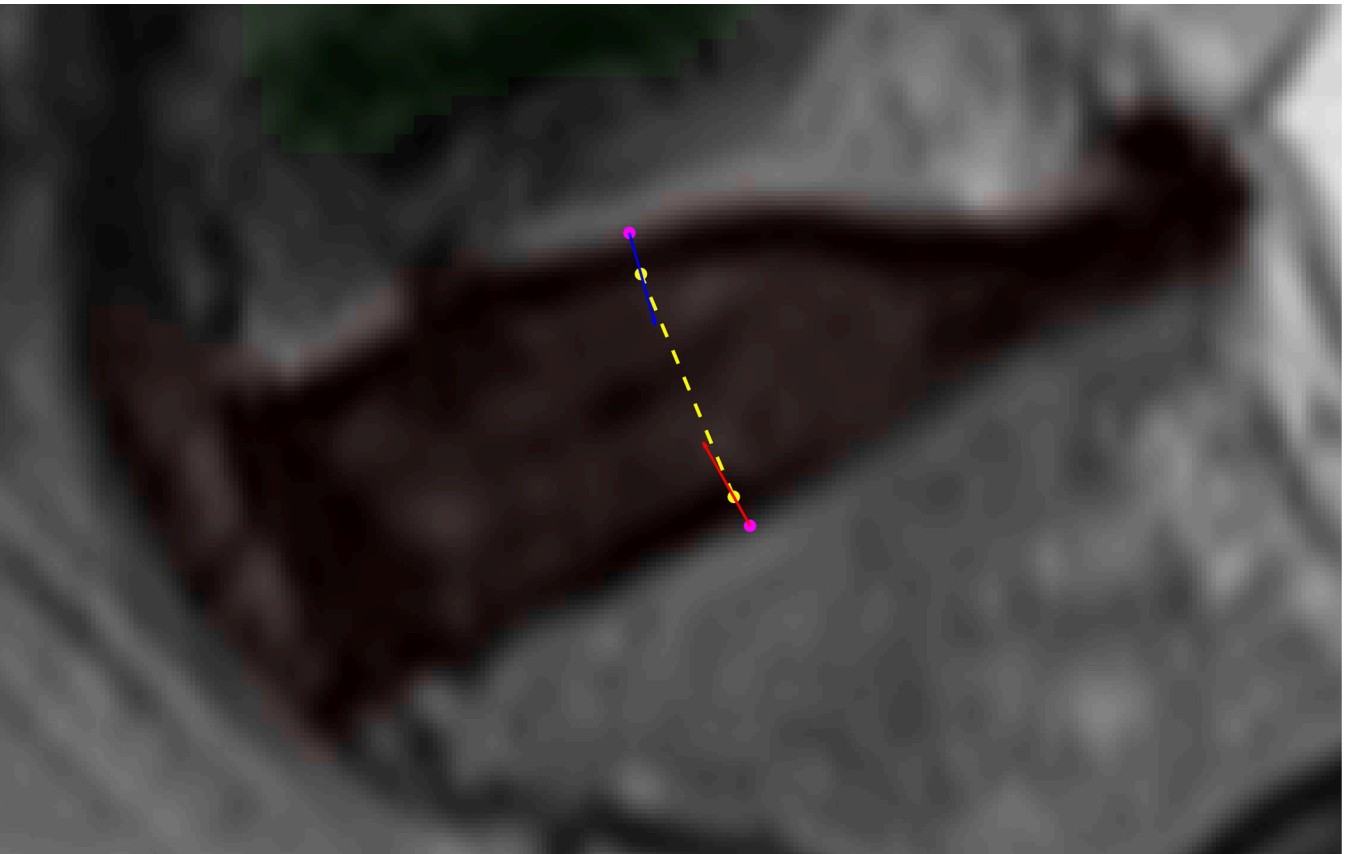

**Fig 11. A visualization of the IVD height measurement stage showing the starting points (magenta), refined points (yellow), final bisecting line segment (yellow dotted line), and the vectors used to shift the points (red and blue arrows).**

respectively. The figure also shows a relatively high misclassification rate of 30% of Pfirrmann Grade 1 IVDs as having Pfirrmann Grade 2. A closer visual inspection of these IVDs found that they indeed have very similar characteristics with Pfirrmann Grade 2, making them visually more difficult to distinguish and hence have a higher chance of misclassification.

The result also shows a few cases (0.6%) where some IVDs having Pfirrmann Grade 2 and 4 were misclassified as having Pfirrmann Grade 5 (rightmost column), as well as a misclassification of 23.8% of IVDs that have Pfirrmann Grade 5 (bottom row). These misclassifications are the result of errors in measuring the IVD height. As we discussed in the previous section, the latter misclassification is attributed to the five outlier cases in which defects in adjacent vertebrae cause the algorithm to predict higher IVD height than the actual one. Currently, we have not devised a way to counter this as that may involve adding an additional layer of expert knowledge to the methodology. However, we envisage two approaches that we can take our work forward in this direction.

First, we could employ an adapted method to the height measurement process to produce a height profile of the IVD instead of the middle height only. This profile could include all the heights measured from the anterior to the posterior of the IVD. This additional information on the height would allow a more representative picture of the IVD's health than the middle height only.

Secondly, we could employ a different approach by considering a shape analysis of the IVD. There are several shape measurements that could provide a better picture of the health of the IVD than just the middle height alone. These include rectangularity, rectilinearity, moments,

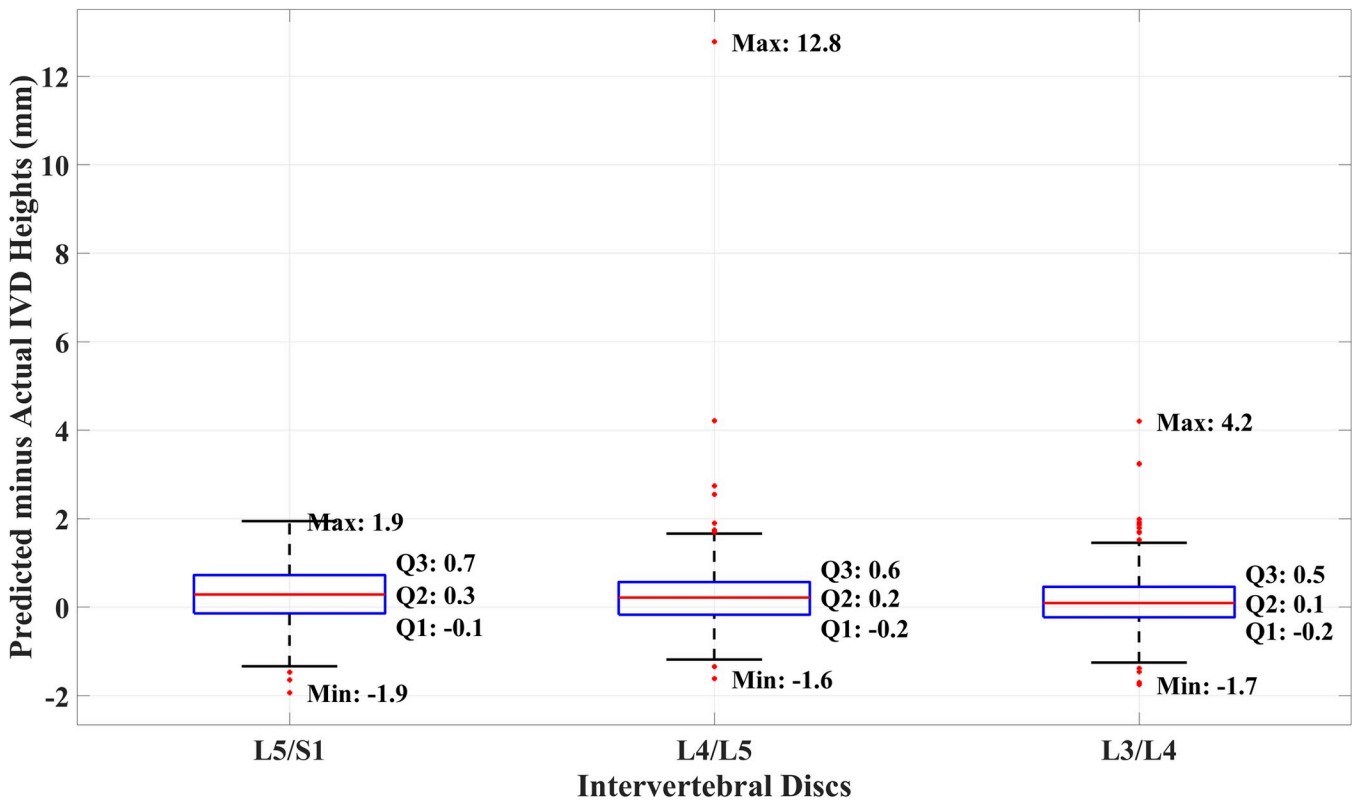

**Fig 12. Box chart of the relative difference between the predicted and actual IVD heights.**

ellipticity [38], and Fourier descriptors [39]. These measurements have been popular in computer vision and used in conjunction with machine learning for shape-based object recognition tasks before the era of deep learning. Alternatively, we could revisit the segmentation process and attempt to include more samples of vertebral defects to allow us to train the model better in distinguishing and separating regions associated with defects in vertebrae from the IVDs. This approach, however, requires acquiring more data that contains the said defects, something which would also be an objective of our future work.

One limitation of this study stems from the imbalanced distribution of the data. As reported in the dataset section, the result of the manual assessment of the last three lumbar IVDs in 515 images is 60, 1002, 167, 295, and 21 cases of Pfirrmann grades 1, 2, 3, 4, and 5, respectively. In terms of percentages, they are 3.9%, 64.9%, 10.8%, 19.0%, and 1.4%, respectively. The relatively very low occurrences of grades 5 in the dataset seem to amplify the shortcomings of the approach in dealing with outlier cases. We also acknowledge another limitation in our study in which we only have one radiologist to perform the initial data analysis and ground truth data preparation. It is widely accepted that inter-observer and intra-observer variances do exist [40] and having more radiologists analyzing the data on multiple occasions can highlight uncertainties in the ground truth data and have an effect on the assessment of the proposed algorithm. Nonetheless, our approach has a significant advantage over the state-of-the-art [9] in that it also produces accurate segmentation of the spinal images and is able to measure the height of the IVDs. Furthermore, as we will show in the following section the methodology can be used to provide descriptive annotation for each IVD which can help explain the rationale behind the classification result produced by the algorithm.

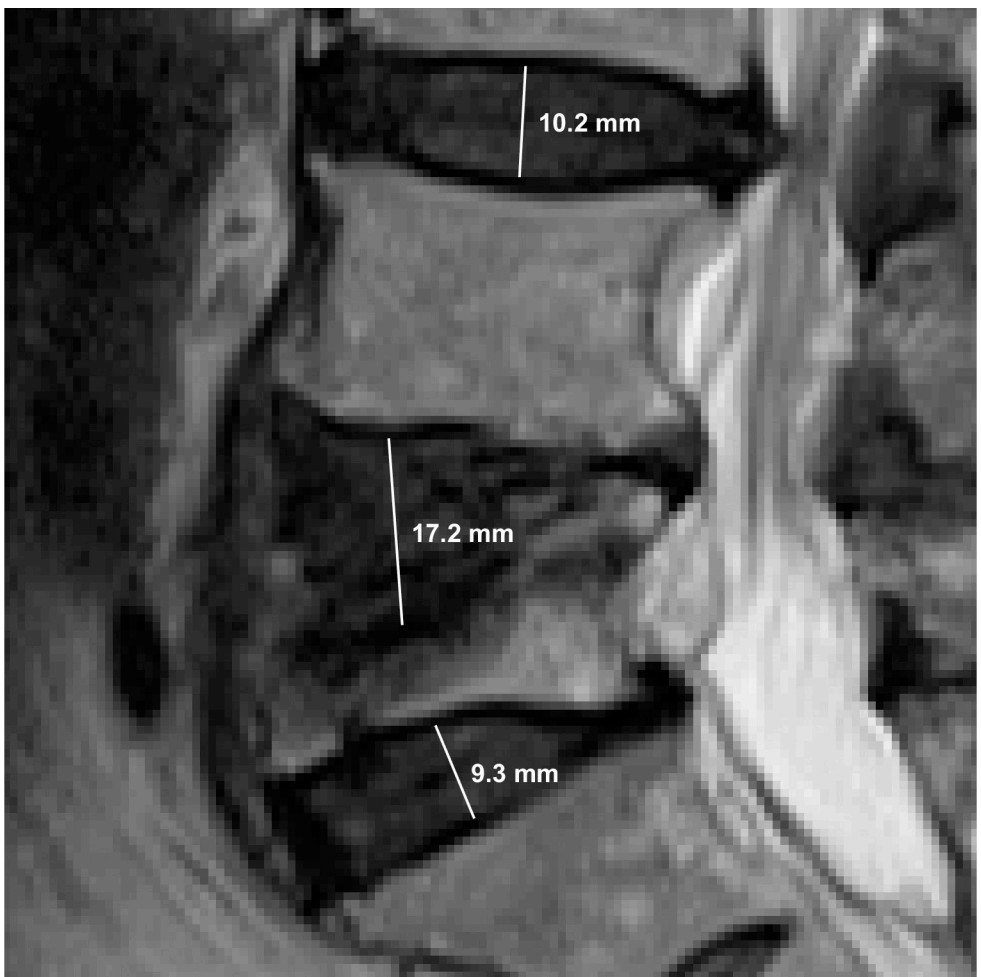

**Fig 13. An example IVD height measurement result that shows the predicted height of three lumbar IVDs.** There is a clear case of trauma at the top plate of the L5 vertebrae causing the algorithm to produce unusually high height prediction for the L4/L5 IVD.

**Table 4. The performance of the Pfirrmann grade classification models.**

| Machine Learning Classifier | Mean Accuracy (%) |
|---|---|
| KNN with Error-Correcting Output Codes | 80.3 |
| SVM with Error-Correcting Output Codes | 85.3 |
| DA with Error-Correcting Output Codes | 78.2 |
| Feedforward Fully Connected Neural Network | 84.5 |
| Decision Tree | 80.1 |
| Ensemble of Decision Trees | 88.1 |

## 3.5. Presentation of annotation results

To finish this section off, we present in Figs 15 and 16, two examples of the annotation results which are produced with the information gathered and predicted at different stages in the process. The presentation crops out a rectangular region surrounding each IVD and marks the border of the detected IVD region with a magenta line. If the IVD nucleus is detected, it will

**Table 5. Range and best hyperparameter values for ensemble binary classification decision tree model.**

| Hyperparameter | Search Range/Set | Best Value |
|---|---|---|
| Method | {RUSBoost [34], AdaBoostM2 [35], Random Forest [36]} | Random Forest |
| Number of Learning Cycles | 10–478 | 15 |
| Minimum Leaf Size | 1–730 | 1 |
| Maximum Number of Splits | 1–1505 | 114 |
| Split Criterion | {deviance, twoing, gdi [37]} | twoing |
| Number Sample Variables | 3–172 | 20 |

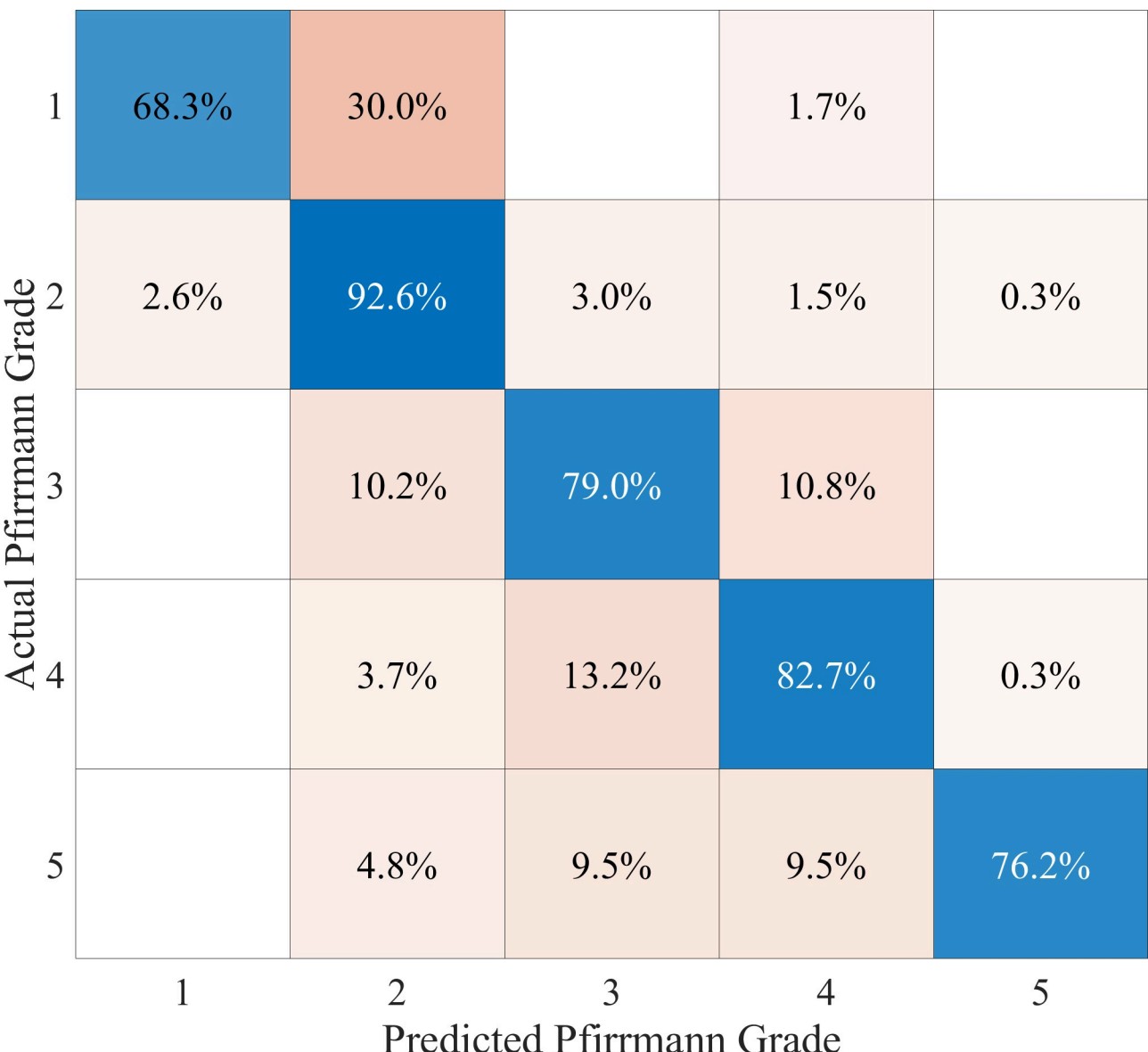

**Fig 14. The confusion matrix of the Pfirrmann grade classification results.** Grades 1 to 4 were produced using the best Feedforward Fully Connected Neural Network classifier whereas grade 5 used the calculated IVD height.

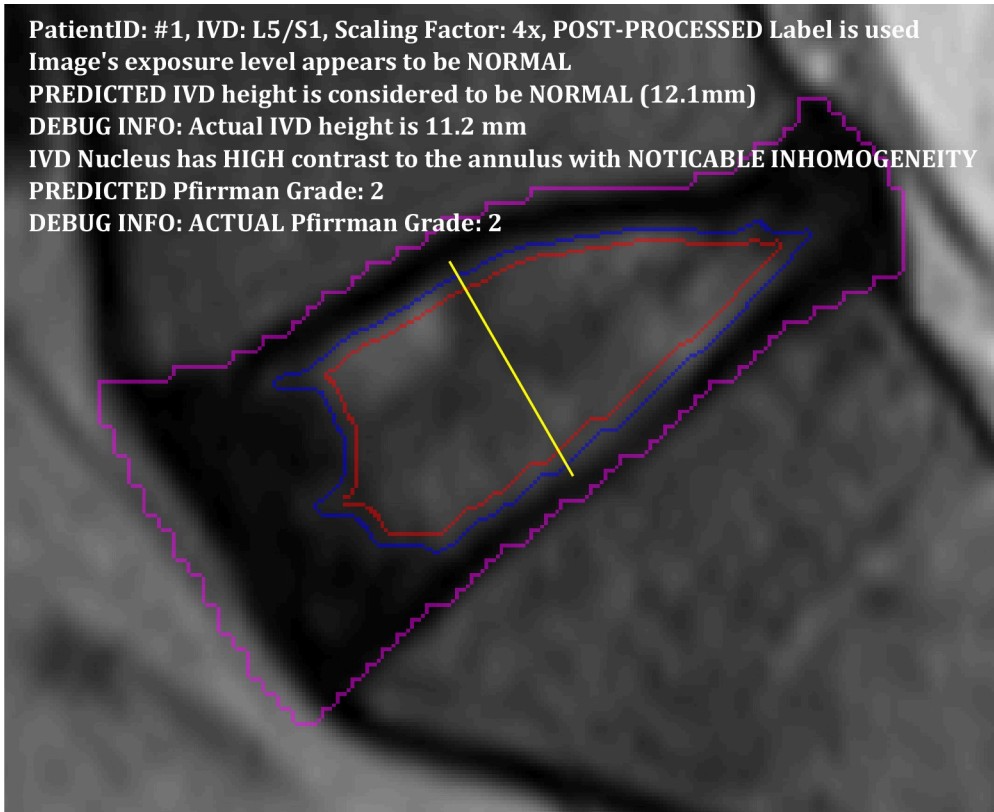

**Fig 15. The annotation of an L5/S1 IVD with reasonably healthy features where the IVD has a normal height and only limited degenerative characteristics.**

mark its border with a blue line and mark the inner nucleus (the inner 90% of the nucleus region) with a red line. A yellow line is used to mark the bisecting line from which the IVD height is calculated.

The first example in Fig 15 shows the annotation of an L5/S1 IVD with reasonably healthy features where the IVD has a normal height and only limited degeneration characteristics. The second example in Fig 16 shows the L4/L5 IVD of the same patient with unhealthy features. This IVD is annotated with a short IVD height and Pfirrmann grade 5. The blue and red lines in this case do not correspond to the nucleus boundary but rather inner region of the IVD that has relatively brighter pixels than the annulus.

## 4. Conclusion

We have presented a methodology to automatically annotate lumbar spine IVDs with their height and Pfirrmann grade that quantifies the degenerative state of the IVD. The method starts with semantic segmentation of a mid-sagittal MRI image into six distinct non-overlapping regions that include the IVD and vertebrae regions. Each IVD region is then located and assigned its label before being analyzed geometrically to find a line segment bisecting the IVD. The Euclidean distance of this line segment is then used as the IVD height. We engineered an image feature, called the self-similar color correlogram, and extracted it from the nucleus of the IVD region as a representation of the region's spatial pixel intensity distribution. We then use the IVD height data and machine learning classification process to predict the Pfirrmann grade of the IVD. We considered five different deep learning networks and six different

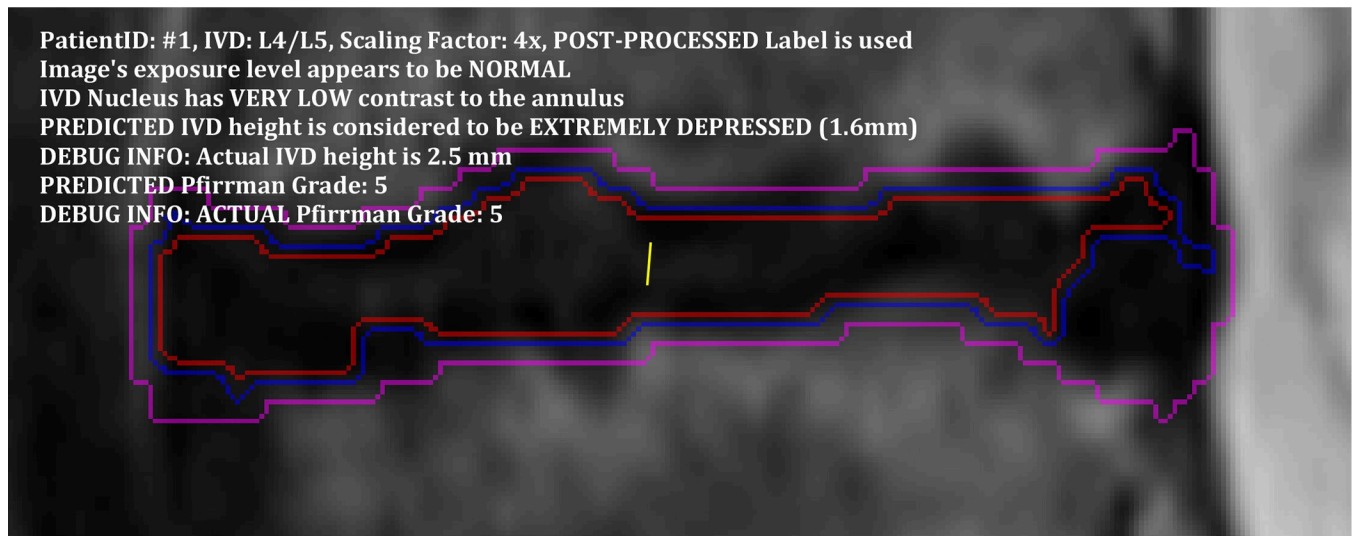

**Fig 16. The annotation of an L4/L5 IVD with unhealthy features.** This IVD is annotated with a short IVD height and Pfirrmann grade 5. The blue and red lines in this case do not correspond to the nucleus boundary but rather inner region of the IVD that has relatively brighter pixels than the annulus.

machine learning algorithms in our experiment and found the ResNet-50 model and Ensemble of Decision Trees classifier to be the combination that gives the best results. Our experiment using a dataset containing 515 MRI studies gives a mean accuracy of 88.1%.

## Author Contributions

**Conceptualization:** Sud Sudirman.

**Data curation:** Daniel Ruslim, Ala Al-Kafri.

**Formal analysis:** Sud Sudirman.

**Funding acquisition:** Friska Natalia.

**Investigation:** Friska Natalia, Sud Sudirman.

**Methodology:** Friska Natalia, Sud Sudirman.

**Project administration:** Friska Natalia.

**Software:** Sud Sudirman.

**Validation:** Daniel Ruslim.

**Writing – original draft:** Ala Al-Kafri.

**Writing – review & editing:** Friska Natalia, Sud Sudirman.

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
