## [Decision Letter · Decision Letter 0]

24 Mar 2024

PONE-D-24-05241Lumbar spine MRI annotation with intervertebral disc height and Pfirrmann grade predictionsPLOS ONE

Dear Dr. Sudirman,

Thank you for submitting your manuscript to PLOS ONE. After careful consideration, we feel that it has merit but does not fully meet PLOS ONE’s publication criteria as it currently stands. Therefore, we invite you to submit a revised version of the manuscript that addresses the points raised during the review process.

We look forward to receiving your revised manuscript.

Kind regards,

Koji Akeda

Academic Editor

PLOS ONE

“This work is supported, in part, by the Ministry of Education, Culture, Research, and Technology of the Republic of Indonesia under grant number: 004-RD-LPPM-UMN/ P-HD/VI/2022.”

“Initials of the authors who received each award: FN

Grant numbers awarded to FN: 004-RD-LPPM-UMN/ P-HD/VI/2022.

The full name of funder: the Ministry of Education, Culture, Research, and Technology of the Republic of Indonesia

URL of funder website: https://www.kemdikbud.go.id/

Did the sponsor or funder play any role in the study design, data collection and analysis, decision to publish, or preparation of the manuscript? No”

Additional Editor Comments:

The manuscript is well written and basically acceptable after minor revision.

Comments:

1. The manuscript should contain an Introduction, Materials and Methods, Results, Discussion, and Conclusion section. Please revise the manuscript.

2. The literature review section should be combined with the introduction section. These two sections are too long. Please summarize the content and shorten the descriptions.

Reviewers' comments:

Reviewer's Responses to Questions

**Comments to the Author**

1. Is the manuscript technically sound, and do the data support the conclusions?

Reviewer #1: Yes

Reviewer #2: Yes

2. Has the statistical analysis been performed appropriately and rigorously? 

Reviewer #1: Yes

Reviewer #2: Yes

3. Have the authors made all data underlying the findings in their manuscript fully available?

Reviewer #1: Yes

Reviewer #2: Yes

4. Is the manuscript presented in an intelligible fashion and written in standard English?

Reviewer #1: Yes

Reviewer #2: Yes

5. Review Comments to the Author

Reviewer #1: We believe this study is useful in the context of the need to rationalize the need for new approaches to increase the efficiency and effectiveness of the radiological diagnostic process. I would like you to continue your research for more accuracy.

Reviewer #2: The authors reported a fascinating application of deep learning in lumbar MRI analysis. By training a deep learning model on a large dataset of annotated MRI images with corresponding Pfirrmann grades, they developed a system that can automatically evaluate and grade intervertebral discs based on their appearance in the images.

As the author pointed out in the discussion section, misclassification of 23.8% of IVDs that have Pfirrmann Grade 5 was a concern of this study (Misclassification of 30% of Pfirrmann Grade 1 IVDs as having Pfirrmann Grade 2 was not a problem because IVDs with both of Pfirrmann Grade 1 and 2 are healthy). Calculating the intervertebral disc height at three points (posterior, anterior, and midline) like "disc height index" might indeed provide a more comprehensive evaluation of disc morphology and potentially enhance the accuracy of the disc height assessment.

I hope that this system will make steady progress and potentially assist radiologists in their diagnosis and treatment planning processes, as well as enable faster and more consistent evaluations across different healthcare settings.

6. PLOS authors have the option to publish the peer review history of their article (what does this mean?). If published, this will include your full peer review and any attached files.

Reviewer #1: No

Reviewer #2: No

---

## [Author Response · Author response to Decision Letter 0]

26 Mar 2024

Our response:

We have reviewed the manuscript again and have made the following changes based on the templates.

• Moved the title down a few lines, remove the bold font style, and reduce the font size to 14.

• We have also changed the way the headings are written to ensure that only the first word is capitalized.

“This work is supported, in part, by the Ministry of Education, Culture, Research, and Technology of the Republic of Indonesia under grant number: 004-RD-LPPM-UMN/ P-HD/VI/2022.”

“Initials of the authors who received each award: FN

Grant numbers awarded to FN: 004-RD-LPPM-UMN/ P-HD/VI/2022.

The full name of funder: the Ministry of Education, Culture, Research, and Technology of the Republic of Indonesia

URL of funder website: https://www.kemdikbud.go.id/

Did the sponsor or funder play any role in the study design, data collection and analysis, decision to publish, or preparation of the manuscript? No”

Our response:

We have removed the Acknowledgement section from the manuscript. The Funding Statement in the online submission form will remain the same.

Our response:

In the last round of the peer-review process, we have already provided a publicly-shared folder on OneDrive where the reviewers can investigate the data that we used. The URL was https://bit.ly/PLOSPFG2024

We have now moved the data to Mendeley Data and can be accessed freely via the following URL: https://data.mendeley.com/datasets/x6ggzp2ycn/1

Our response:

We have checked the references to ensure that they are correctly cited and none of them had been retracted. We made the following changes:

• Corrected reference [1] and [26] items which were incorrectly referred to as [Internet] because the reference manager has a URL in their record.

• Updated the URL of reference [4] because the old one has changed.

• Updated the last accessed date of all web page reference items [3, 4, 10, 17] to 25 March 2024

• Replaced reference item [30] with a new one due to the article’s recent retraction.

The previous one was:

30. Chugh H, Gupta S, Garg M, Gupta D, Juneja S, Turabieh H, Na Y, Kiros Bitsue Z, Others. Image retrieval using different distance methods and color difference histogram descriptor for human healthcare. J Healthc Eng. 2022;2022.

The new one is:

30. Hee-Hyung BuNam-Chul Kim Byoung-Ju Yun, Kim S-H. Content-Based Image Retrieval Using Multi-Resolution Multi-Direction Filtering-Based CLBP Texture Features and Color Autocorrelogram Features. J Inf Process Syst. 2020;16(4):991–1000.

This change does not change or invalidate the statement in the manuscript in which the article was referenced. It was used to back up the argument that states “The majority of information in a color correlogram is concentrated along the main diagonal cells of the matrix hence several studies have suggested not to use the entire matrix but only these cells [30,31]”. The replacement article also supports the same argument.

Additional Editor Comments:

The manuscript is well written and basically acceptable after minor revision.

Comments:

1. The manuscript should contain an Introduction, Materials and Methods, Results, Discussion, and Conclusion section. Please revise the manuscript.

2. The literature review section should be combined with the introduction section. These two sections are too long. Please summarize the content and shorten the descriptions.

Our response:

Thank you for the comments. We have reviewed the structure of our manuscript and made the following changes:

• Combined the Introduction section and the Literature Review section into one section, and shorten the result by ~20%. 

• Replaced the title of the Methodology section to Materials and Methods, and 

• Replaced the title of the Experimental results and analysis section to Results and discussion.

Reviewers' comments:

Reviewer's Responses to Questions

Comments to the Author

1. Is the manuscript technically sound, and do the data support the conclusions?

Reviewer #1: Yes

Reviewer #2: Yes

2. Has the statistical analysis been performed appropriately and rigorously? 

Reviewer #1: Yes

Reviewer #2: Yes

3. Have the authors made all data underlying the findings in their manuscript fully available?

Reviewer #1: Yes

Reviewer #2: Yes

4. Is the manuscript presented in an intelligible fashion and written in standard English?

Reviewer #1: Yes

Reviewer #2: Yes

5. Review Comments to the Author

Reviewer #1: We believe this study is useful in the context of the need to rationalize the need for new approaches to increase the efficiency and effectiveness of the radiological diagnostic process. I would like you to continue your research for more accuracy.

Reviewer #2: The authors reported a fascinating application of deep learning in lumbar MRI analysis. By training a deep learning model on a large dataset of annotated MRI images with corresponding Pfirrmann grades, they developed a system that can automatically evaluate and grade intervertebral discs based on their appearance in the images.

As the author pointed out in the discussion section, misclassification of 23.8% of IVDs that have Pfirrmann Grade 5 was a concern of this study (Misclassification of 30% of Pfirrmann Grade 1 IVDs as having Pfirrmann Grade 2 was not a problem because IVDs with both of Pfirrmann Grade 1 and 2 are healthy). Calculating the intervertebral disc height at three points (posterior, anterior, and midline) like "disc height index" might indeed provide a more comprehensive evaluation of disc morphology and potentially enhance the accuracy of the disc height assessment.

I hope that this system will make steady progress and potentially assist radiologists in their diagnosis and treatment planning processes, as well as enable faster and more consistent evaluations across different healthcare settings.

Our response:

Thank you for your time and effort to review our paper. We are glad that both of you found our work to be useful and fascinating. The team hopes that we can build on this work and implement the solution more widely and increase its positive impact to society.

---

## [Editor Report · Decision Letter 1]

27 Mar 2024

Lumbar spine MRI annotation with intervertebral disc height and Pfirrmann grade predictions

PONE-D-24-05241R1

Dear Dr. Sudirman,

We’re pleased to inform you that your manuscript has been judged scientifically suitable for publication and will be formally accepted for publication once it meets all outstanding technical requirements.

Kind regards,

Koji Akeda

Academic Editor

PLOS ONE

Additional Editor Comments (optional):

This paper is a significant contribution and potentially be suitable for publication in PlosOne.
---

## [Editor Report · Acceptance letter]

29 Apr 2024

PONE-D-24-05241R1 

PLOS ONE

Dear Dr. Sudirman, 

I'm pleased to inform you that your manuscript has been deemed suitable for publication in PLOS ONE. Congratulations! Your manuscript is now being handed over to our production team.

Kind regards, 

on behalf of

Dr. Koji Akeda 

Academic Editor

PLOS ONE